

# Peripheral-physiological and neural correlates of the flow experience while playing video games: a comprehensive review

Shiva Khoshnoud, Federico Alvarez Igarzábal and Marc Wittmann

Institute for Frontier Areas of Psychology and Mental Health, Freiburg, Germany

## ABSTRACT

The flow state is defined by intense involvement in an activity with high degrees of concentration and focused attention accompanied by a sense of pleasure. Video games are effective tools for inducing flow, and keeping players in this state is considered to be one of the central goals of game design. Many studies have focused on the underlying physiological and neural mechanisms of flow. Results are inconsistent when describing a unified mechanism underlying this mental state. This paper provides a comprehensive review of the physiological and neural correlates of flow and explains the relationship between the reported physiological and neural markers of the flow experience. Despite the heterogeneous results, it seems possible to establish associations between reported markers and the cognitive and experiential aspects of flow, particularly regarding arousal, attention control, reward processing, automaticity, and self-referential processing.

## INTRODUCTION

What makes everyday experiences genuinely satisfying? *Csikszentmihalyi (1975)* introduced the concept of "flow" or "being in the zone" as an optimal state in which complete absorption in an activity is reached and is accompanied by a sense of enjoyment and apparent ease of processing. Flow is a subjective experience in which actions seem to happen effortlessly, fluently, and almost automatically. According to this theory, the clearest sign of flow is the merging of action and awareness in a way that "a person in flow is aware of his actions but not of the awareness itself" (*Csikszentmihalyi, 1975*, pp 38). Paradigmatic examples of flow-inducing activities include the artist who is completely immersed in the activity of creating a work of art or playing an instrument and the athlete or game player who follows clear goals and perceives a match between demands and skills. The following factors enable the flow experience, which in combination create a deep sense of enjoyment (*Csikszentmihalyi, 1975*; *Csikszentmihalyi, 1990*; *Jackson & Eklund, 2004*): (1) the balance between the level of skill and the challenges of the task, (2) clear goals of the activity, (3) clear immediate feedback of action results, (4) merging of action and awareness, (5) high concentration, (6) a sense of control over the activity, (7) a loss of

Corresponding author
Shiva Khoshnoud,
khoshnoud@igpp.de

self-awareness, (8) a loss of the sense of time, and (9) autotelic experience or an intrinsically rewarding perception of the activity.

Among the nine key components of flow, the first three (balance between skills and challenges, clear goals, and immediate feedback) can also be understood as antecedents to or activity-related preconditions for flow; the remaining six items are characteristics of this subjective state. An optimal skill-challenge balance is considered to be the main antecedent which facilitates entering into the flow state (*Csikszentmihalyi & Csikszentmihalyi, 1992*; *Engeser & Rheinberg, 2008*; *Fong, Zaleski & Leach, 2015*; *Keller & Blomann, 2008*). If the challenge level of the activity outweighs the skill level of the person performing it, the person will become frustrated. In contrast, if the challenge level is lower than the skill level, the person will become bored (the flow-channel model; *Csikszentmihalyi, 1975*; *Csikszentmihalyi, 1990*). Although the skill-challenge balance is a prerequisite for the flow experience, it does not guarantee entering into the flow state. *Fong, Zaleski & Leach (2015)* showed that additional variables, such as age, cultural characteristics, domain of application (leisure or work/educational contexts), and methodology (how the skill-challenge balance is evaluated) may distinctively influence the relationship between flow and the skill-challenge balance. *Engeser & Rheinberg (2008)* showed that flow was still high even when the demand was low in activities that were subjectively evaluated as important to succeed in. Investigation of the moderating impact of personality characteristics, such as the action-state orientation, revealed that individuals with a strong habitual action orientation are more sensitive to modulations of the skill-challenge balance (*Keller & Bless, 2008*). The likelihood of the ensuing flow experience can also be altered by personality factors. A study by Ullén and colleagues (*2012*) reported a negative correlation between flow proneness (understood as the individual propensity to experience flow) and neuroticism. *De Manzano et al. (2013)* suggested that lower trait impulsivity could facilitate the propensity to experience flow.

Despite such potential modulators, the relationship between the skill-challenge balance and flow postulated in the flow-channel model affords an approach for experimentally manipulating flow by presenting different levels of challenge. Although inducing flow under controlled laboratory conditions has been considered difficult, game-based paradigms are especially promising, as: (a) games offer challenging tasks that require training skills and provide clear goals, as well as immediate feedback (*Alvarez Igarzábal, 2019*; *Salen & Zimmerman, 2003*), and (b) the difficulty level of a game can relatively easily be manipulated to achieve the skill-challenge balance. An important driver of enjoyment in games comes from *effectance motivation*, a term coined by *Nacke (2012)*, which is the feeling of empowerment in players when they see the impact of their actions. This feeling of control can be experienced when the game's challenge matches the player's skills and goals and immediate feedback are provided. The experimental approaches include dynamic matching of the game's difficulty level to the player's skill level (*Keller & Bless, 2008*; *Rheinberg & Vollmeyer, 2003*) or pre-testing the participants' skills to individually assign appropriate challenge levels in the game (*Moller et al., 2007*). This can help to contrast three experimental conditions: easy, optimal, and overwhelming. *Rheinberg & Vollmeyer (2003)* evaluated the impact of modulating the task difficulty on the flow experience in

two different video games, *Roboguard* and *Pac-Man*. The highest level of flow was reported following those trials in which the game's difficulty was set to a medium level rather than to a low or high level. A self-selected level of difficulty (autonomy) was also suggested to be an important determinant of flow (*De Sampaio Barros et al., 2018*; *Moller, Meier & Wall, 2010*). Some studies implemented an additional immersion condition, which was described as the gradual process of transporting the player's mind into the virtual world by manipulating factors like graphics, sound, and gameplay (*Drachen et al., 2010*; *Nacke, Stellmach & Lindley, 2011*). Immersion is an important concept in virtual reality (VR) and computer-gaming research and seems closely connected to flow in these contexts, as both pertain to a pleasant, absorbed mental state common in gamers. However, flow and immersion can be distinguished due to subtle structural differences (for further information, please see *Michailidis, Balaguer-Ballester & He, 2018*).

Experimental flow research using computer games has important implications, such as understanding peak performance. Flow and performance seem to be closely related (*Csikszentmihalyi, Abuhamdeh & Nakamura, 2005*; *Engeser & Rheinberg, 2008*; *Jin, 2012*; *Keller & Bless, 2008*; *Landhäußer & Keller, 2012*). High performance levels are typically expected during the experience of flow because frustration and boredom lead to diminished concentration and, consequently, to poor performance. It is still unclear whether flow influences performance or vice versa (*De Kock, 2014*; *Landhäußer & Keller, 2012*). *Engeser & Rheinberg (2008)* found that the flow experience led to improved performance in participants who played the game *Pac-Man* at three difficulty levels, while *Jin (2012)* reported that successful performance resulted in a greater flow experience in participants who played the games *Call of Duty: World at War* and *Trauma Center: New Blood*. Therapeutic settings might also benefit from experimental flow research utilizing video games. The flow state is an experiential feature of altered states of consciousness which can lead to a diminished sense of self and time (*Wittmann, 2015*; *Wittmann, 2018*). Distortions in the notions of the self and time have been reported in many patient groups with psychiatric disorders (*Khoshnoud et al., 2017*; *Vogel et al., 2019*; *Vogel et al., 2018*). The sense of self and time is overly represented in individuals with anxiety and depression, who are stuck with themselves in time and experience states that are the complete opposite of flow (*Liknaitzky, 2017*). Video games specifically make time fly in a pleasant way, which is one of the main aspects of the flow experience (*Bisson, Tobin & Grondin, 2012*; *Tobin, Bisson & Grondin, 2010*). The induction of flow states has been shown to alter the sense of time (*Sinnett et al., 2020*). By measuring flow levels and the temporal-processing ability, Sinnett and colleagues identified that the higher the subjective flow experience of the sport or music performance, the better the participant performed in the post-performance timing task compared to the pre-performance timing task. Inducing flow states in these individuals could potentially lower symptoms of anxiety and depression. A study with the video game *Boson X* reported that playing it for six weeks reduced self-rumination and enhanced cognitive capacities in individuals with depression (*Kühn et al., 2018*). Considering the beneficial nature of flow on psychiatric symptoms, creating a flow experience might provide a helpful remedy in clinical psychology.[1] Since flow is an enjoyable mental state, keeping the player in a flow state is considered to be one of the most important goals

[1] This is the approach in the EU-funded project VIRTUALTIMES - Exploring and modifying the sense of time in virtual environments under the principal investigators Kai Vogeley (Cologne), Anne Giersch (Strasbourg), Marc Erich Latoschik, Jean-Luc Lugrin (Würzburg), Giulio Jacucci, Niklas Ravaja (Helsinki), Marc Wittmann (Freiburg) and Xavier Palomer, Xavier Oromi (Barcelona).

for game designers (*Chen, 2006*; *Salen & Zimmerman, 2003*; *Schell, 2008*). To address this, game designers and researchers attempt to maintain the player's flow state through affect-based, dynamic difficulty-adjustment (DDA) techniques (*Afergan et al., 2014*; *Liu et al., 2009*; *Park, Sim & Lee, 2014*).

To successfully examine research questions related to flow and performance, therapeutic use, or gamer experience, the experimental paradigm must be accompanied by a valid measurement of the flow state. The concept of flow was initially investigated using the Experience Sampling Method (ESM) in naturalistic contexts (*Csikszentmihalyi & Csikszentmihalyi, 1992*). This method involves contacting participants at random moments throughout the day and asking them questions about the nature and quality of their experience (*Csikszentmihalyi & Larson, 1983*). Later studies optimized the ESM and designed new questionnaires to evaluate the flow experience. Some of these are the Flow State Scale (FSS, *Jackson & Marsh, 1996*) first designed in the context of sports, the Flow Short Scale *(Flow-Kurzskala*; FKS, *Rheinberg & Vollmeyer, 2003*) developed for different fields of activity, the Flow Questionnaire (FQ, *Sherry et al., 2006*), and the flow subscale of the Game Experience Questionnaire (GEQ, *IJsselsteijn et al., 2007*) designed for evaluation of the subjective gaming experience, the Virtual-Course Flow Measure (*Shin, 2006*) developed in the context of online learning, the Flow State Scale for Occupational Tasks (*Yoshida et al., 2013*), and the Work-Related Flow Inventory (WOLF, *Bakker, 2008*) specifically aimed at measuring the flow of employees. However, assessment of the flow experience with these retrospective questionnaires interrupts the ongoing activity and probably disrupts the flow experience. Utilizing these self-reported, post-task questionnaires cannot provide information about characteristics of this experience, like the mean duration or depth of flow. It is crucial to find non-disruptive, objective measures to continuously evaluate the flow experience. One way to assess this experience without interrupting it is to find neural and electrophysiological correlates of this state, which in turn could help to better understand the underlying physiological mechanisms. Apart from these internal correlates, another promising method would be the application of dual-task approaches that indirectly measure the extent to which subjects experience flow by assessing their levels of focused attention.

Here we provide an overview of all findings concerning the physiological and neural correlates of the flow experience. We believe this is the first review that combines both the physiological and neural correlates of flow in the context of video games. *Harris, Vine & Wilson (2017)* conducted a review on the neurocognitive mechanisms of flow with more emphasis on sports, as well as on the role of attention, suggesting attentional changes as the fundamental mechanism for the creation of the flow state. The role of physiological arousal was not discussed in detail in their review. In a recent systematic review conducted by *Knierim et al. (2018)*, only peripheral-nervous-system indicators of flow were explored in a broad range of tasks identifying increased levels of arousal as a central approach to the physiological measurement of flow. We aimed to make a key contribution to the field of flow in the context of video games by considering reflections of flow in the central and peripheral nervous systems and integrating main physiological and neural mechanisms of the flow experience. Based on the results of the reviewed studies, we suggest possible explanations

for the heterogeneity of the findings and how to distinguish the internal phenomenon of flow from the external characteristics of the task. For the sake of consistency, given that different studies label conditions in different ways, in the following we will refer to difficult conditions as "over-challenged", to easy ones as "under-challenged", and to optimally challenged conditions as "flow."

## SURVEY METHODOLOGY

We identified relevant academic articles in the Web of Science and PubMed databases using the search terms: ((flow OR absorption) AND (physiological OR electrophysiological OR neurophysiology OR brain activity OR neural activity) AND (game OR video game OR gameplay)). This search provided a total of 215 citations. After removing duplicates, the set of 55 articles comprising peer reviewed, empirical studies focused on the physiological and neural phenomenon of the flow experience while playing video games without a-priori, publication-date restrictions was selected. Another 18 studies were identified by scanning the references listed in the literature found in the initial search. Thirty-five articles, including research on exergames (videogames in the form of exercise) and multiplayer games, were further excluded by applying the inclusion criteria to the full texts of the manuscripts, as they introduced other confounding factors (e.g., movement effects and social interaction). The 38 articles included in the review are presented in two sections: peripheral-physiological and neural correlates of flow.

### Peripheral-physiological correlates of flow

Before introducing the empirical work, it is necessary to discuss the theoretical background of the peripheral-physiological correlates of flow. During the flow state, feelings of enjoyment along with high levels of concentration and focused attention are indicative of the involvement of the emotional and attentional systems of the brain. Based on this line of thought, several hypotheses have been proposed, which we will discuss in the following sections. First, the experimental approach by *Kivikangas (2006)* defined the flow experience as a state of positive valence and heightened arousal. Later, de Manzano and colleagues described the physiology of flow as a combination of positive valence, heightened arousal, and effortless attention that arises through the interaction between positive affect and focused attention (*De Manzano et al., 2010*; *Ulrich, Keller & Grön, 2016*). According to this hypothesis, flow is associated with parasympathetic modulation of the sympathetic branch of the autonomic nervous system (ANS). *Ullén et al. (2010)* argued that this co-activation of the sympathetic nervous system (SNS) and the parasympathetic nervous system (PNS) acts as a physiological coping mechanism for high demands of attention and can distinguish between states of effortful and effortless attention. In contrast to the notion of effortless attention, which is typical of the flow experience, Keller and colleagues (*2011*) argued that flow involves considerable mental effort due to a high degree of involvement along with the challenging nature of the task.

Finally, by combining the stress-model with the flow-model, *Peifer et al. (2014)* proposed that the experience of challenge during flow induces a certain amount of stress accompanied by heightened physiological arousal, as indicated by increased activation of the SNS (fast

response system) and the hypothalamic–pituitary–adrenal (HPA) axis (slow response system). They suggested an inverted U-shaped relationship between flow experience and physiological arousal with a moderate arousal level during flow and lower and higher levels of arousal for the under-challenged and over-challenged conditions, respectively. Reported negative effects of exogenous cortisol dosage on experienced flow supports their recent proposition of an inverted U-shaped relationship between cortisol and flow (*Peifer et al., 2015*). Empirical studies in this field (see Table 1) are presented in the following section. However, inconsistent results during flow states show that the relationship among the flow experience, arousal, and mental effort is highly dependent on the task.

## Positive valence and heightened physiological arousal

One of the first studies that investigated correlations among valence, arousal, and flow was conducted by *Kivikangas (2006)*. The study assessed the participants' facial electromyographic (EMG) activity as an index of emotional valence (*Lang et al., 1993*; *Larsen et al., 2008*) and electrodermal activity (EDA) as a sensitive measure of arousal (*Boucsein, 2012*) while they played the science-fiction computer game *Halo: Combat Evolved*. The activity of the corrugator supercilii muscle (CS, "frowning muscle"), an index of negative valence, was negatively associated with the flow scores assessed by the FSS questionnaire, showing decreased negative valence during the experience of flow. No significant effect was found for zygomaticus-major (ZM, "smiling muscle") or for orbicularis oculi-muscle (OO, "eyelid muscle") activities - indices of positive valence –or for EDA activity. *Chanel et al. (2008)* employed physiological measures, including cardiovascular activity and EDA, to determine the three emotional states induced by under-challenged, flow, and over-challenged conditions of the game *Tetris* by modulating the difficulty of the game. Cardiovascular activity is reflective of ANS activity, with heart rate (HR) being stimulated by the SNS and inhibited by the PNS activity (*Shaffer & Ginsberg, 2017*). The behavioral results demonstrated that participants felt the highest positive valence and had a medium arousal level in the flow condition in contrast to the under-challenged and the over-challenged conditions. Electrophysiological measures showed heightened arousal levels as the difficulty of the game increased, which was identified by an increase in EDA and the HR (increased SNS activity). Utilizing a modified version of the first-person-shooter (FPS) game *Half-Life 2* with specifically-designed levels, *Nacke & Lindley (2008)* assessed immersion along with under-challenged and flow states by correlating these states with the objective electrophysiological measures. The significantly highest mean ZM muscle activity and EDA values were detected during the flow game level. However, the flow experience was not evaluated by any questionnaire, and it is not clear whether participants experienced flow in the flow condition of the game. In a correlational study employing three FPS games, no significant correlation was reported between EDA activity and flow scores assessed by the flow dimension of the GEQ, while the HR was reported to negatively correlate with flow (*Drachen et al., 2010*).

*Bian et al. (2016)* presented a physiological evaluation model for the state of flow in the virtual-reality (VR) game *Air Bombardment*. In contrast to the findings of the study by *Nacke & Lindley (2008)*, the authors reported no correlations between ZM activity and

Khoshnoud et al. (2020), *PeerJ*, DOI 10.7717/peerj.10520

**Table 1** Studies on peripheral-physiological correlates of the flow state.

| Ref. | N subjects | Age mean/ range | Sample | Experiment type | Design | Measure | Peripheral-physiological correlates of flow |
|------|-----------|-----------------|--------|-----------------|--------|---------|---------------------------------------------|
| *Kivikangas (2006)* | 32(m) | 17–34 | Healthy students | FPS game: Halo: combat evolved | 40 min gameplay | FEMG, EDA | Negative correlations between CS activity & flow |
| *Chanel et al. (2008)* | 20 (7f) | 27 | Healthy participants | Tetris game | 5 min Under-challenged, Flow, Over-challenged | EDA, BP, Res, T | Increase in EDA & HR with increasing difficulty |
| *Nacke & Lindley (2008)* | 25(m) | 19–38 | Healthy students | FPS game: Half-Life 2 | 10 min Under-challenged, Immersion, Flow | FEMG, EDA | Highest ZM activity & EDA values in the flow condition |
| *Drachen et al. (2010)* | 16 | – | – | FPS games: Prey, Doom3, Bio-chock | 20 min gameplay | EDA, HR | Negative correlation between HR & flow subscale score |
| *Chanel et al. (2011)* | 20(7f) | 27 | Healthy participants | Tetris game | 5 min Under-challenged, Flow, Over-challenged | EDA, BP, Res, T, EEG | Least LF-HRV in the flow condition |
| *Keller et al. (2011)* | 8(4f) 61(m) | – | Healthy students | Computerized knowledge task, Tetris game | Under-challenged, Flow, Over-challenged | HRV, Cortisol | Lower HRV & higher cortisol in the flow condition |
| *Peifer et al. (2014)* | 22(m) | 20–34 | Healthy students | Cabin Air Management System software | 60 min performance | ECG, Cortisol | Inverted U-shaped relationship of LF-HRV & cortisol level with the flow experience; positive linear relationship between HF-HRV & flow |
| *Léger et al. (2014)* | 36 | – | Healthy students | Enterprise Resource Planning software | Under-challenged, Flow, Over-challenged | ECG, EDA, EEG | Smaller variation of EDA, lower HR, & higher HRV in the flow condition |
| *Harmat et al. (2015)* | 77(40f) | 27 | Healthy subjects | Tetris game | 6 min Under-challenged, Flow, Over-challenged | fNIRS, ECG, Res | Larger RD & lower LF-HRV in the flow condition |

Khoshnoud et al. (2020), *PeerJ*, DOI 10.7717/peerj.10520

**Table 1** (*continued*)

| Ref. | N subjects | Age mean/range | Sample | Experiment type | Design | Measure | Peripheral-physiological correlates of flow |
|---|---|---|---|---|---|---|---|
| *Tozman et al. (2015)* | 18(6f) | 19 | Healthy students | Sporting race game: Rfactor | 6 min Under-challenged, Flow, Over-challenged | ECG | Negative linear relationship between LF-HRV & flow in the flow condition; Inverted U-shaped relation between LF-HRV / HF-HRV & flow in the anxiety condition |
| *Bian et al. (2016)* | 36(16f) | 20–27 | Healthy adults | VR game: Air Bombardment | 6 min gameplay | ECG, Res, FEMG | Inverted U-shaped relationship between LF-HRV, HF-HRV, & flow |
| *Harris, Vine & Wilson (2016)* | 33(10f) | 20 | Healthy students | Simulated car racing game | Under-challenged, Flow, Over-challenged | ECG, eye gaze position | Lower SD of horizontal gaze position & Lower HF-HRV in the flow condition |
| *Ulrich, Keller & Grön (2016)* | 23(m) | 24 | Healthy students | Mental arithmetic task | 30 s Under-challenged, Flow, Over-challenged | fMRI, EDA | Greater EDA in the flow condition |
| *Tian et al. (2017)* | 40(27f) | 17–24 | Healthy students | Blocmania 3D game | 6 min Under-challenged, Flow, Over-challenged | ECG, Res, EDA | Faster respiratory rate, increased RD, moderate HR, moderate HRV, & moderate EDA in the flow condition |
| *De Sampaio Barros et al. (2018)* | 20 (7f) | 26 | Healthy adults | Tetris, Pong games | 3 min Under-challenged, Flow, Over-challenged, Autonomy | ECG, Res, NIRS | Lower HRV in the autonomy condition |
| *Kozhevnikov et al. (2018)* | 56(17f) | – | Healthy students | Unreal Tournament 2004 game | 30 min Under-challenged, Flow, Over-challenged | ECG | Lower HF-HRV in the flow condition |
| *Moreno et al. (2020)* | 1 | 27 | Expert gamer | Portal game | 45 min gameplay | EDA, EEG | Increased EDA during moments of goal attainment |

Notes.

ECG, electrocardiography; EEG, electroencephalography; FEMG, facial electromyography; EDA, electrodermal activity; BP, blood pressure; T, temperature; HR, heart rate; HP, heart period; ZM, zygomaticus major; CS, corrugator supercilii; HRV, heart rate variability; LF-HRV, low-frequency heart-rate variability; HF-HRV, high-frequency heart-rate variability; Res, respiration; RD, respiratory depth; fNIRS, functional near-infrared spectroscopy; NIRS, near-infrared spectroscopy; fMRI, functional magnetic resonance imaging; FPS, first person shooter; SD, standard deviation; m, male; f, female.

flow scores as assessed by the FKS questionnaire. Using a mental-arithmetic task, a study by Ulrich and colleagues found an inverse U-shaped pattern for EDA with significantly higher values during the flow condition than during the under-challenged and over-challenged conditions, highlighting higher arousal levels during the experience of flow (*Ulrich, Keller & Grön, 2016*). EDA was also assessed while playing the *Blocmania 3D* game with three levels of difficulty corresponding to under-challenged, flow, and over-challenged states (*Tian et al., 2017*). Difficulty manipulation was assessed with the FSS questionnaire. The highest flow state which was reported during the flow condition of the game was associated with moderate EDA activity, which reflects moderate sympathetic arousal. In a single-case study, *Moreno et al. (2020)* reported that a flow-like state in an expert gamer while playing the puzzle game *Portal* coincided with increased EDA. In their study, physiological assessment was not conducted during the moments of flow, but when individuals were goal-oriented during gameplay.

## Co-activation of the sympathetic and the parasympathetic nervous systems

Findings regarding the relationship between the flow state and ANS are mixed, as both sympathetic and parasympathetic activity have been shown to correlate with flow in combination and alone. *De Manzano et al. (2010)* argued that the flow state experienced while playing piano is linked to increased parasympathetic modulation of sympathetic activity. Their study showed that the flow reports of the pianists (as assessed by the FSS questionnaire) correlated with a decreased heart period (HP, increased SNS activity), increased heart rate variability (HRV, fluctuations in the time intervals between heartbeats and an index of parasympathetic activity) (*Laborde, Mosley & Thayer, 2017*), an increased LF/HF ratio (low frequency HRV/high frequency HRV, reflecting autonomic balance between the SNS and the PNS), and increased respiratory depth (RD, increased PNS activity) (*De Manzano et al., 2010*). *Chanel et al. (2011)*, in contrast, reported less low-frequency heart-rate variability (LF-HRV) during flow compared to under-challenged and over-challenged conditions while playing *Tetris*. In a computerized knowledge task, *Keller et al. (2011)* reported lower HRV during the flow condition compared to under-challenged and over-challenged conditions, indicating less parasympathetic activity. A more detailed assessment of cardiovascular and respiratory responses was performed by *Harmat et al. (2015)* during trials of *Tetris* gameplay in all three conditions: under-challenged, flow, and over-challenged. The flow condition was characterized by the highest levels of flow measured by the FSS questionnaire. More flow was associated with greater respiratory depth (reflecting increased parasympathetic activity) and lower LF-HRV (reflecting both sympathetic and parasympathetic influences). Given the lack of a significant relation between HF-HRV (high-frequency HRV, a direct measure of parasympathetic activation) and flow, their results did not clearly support the hypothesis that the flow state is linked to activation of the sympathetic and the parasympathetic nervous systems (*Harmat et al., 2015*). Tian and colleagues (2017) also reported moderate HR and HRV along with increased RD during the flow condition while playing the game, suggesting increased parasympathetic modulation of sympathetic activity during the flow experience. In a FPS

game called *Unreal Tournament 2004*, lower HF-HRV was reported in subjects playing the game during the flow condition compared to under-challenged and over-challenged conditions (*Kozhevnikov et al., 2018*). Given the lack of a significant change in the LF-HRV values, the authors argued that this pattern of reduction in parasympathetic activity is critical for achieving flow. The flow experience was not assessed directly in their investigation, and it is not clear whether subjects subjectively experienced higher flow while playing the game during the flow condition as compared to the other two conditions.

Both the HRV and the HF-HRV are considered as sensitive indices of parasympathetic activity (*Laborde, Mosley & Thayer, 2017*; *Malik et al., 1996*; *Shaffer & Ginsberg, 2017*), which was reported to be causally involved in flow experience (*Colzato, Wolters & Peifer, 2018*). The interpretation of LF-HRV is controversial, since it is considered a marker of sympathetic modulation (*Kamath & F, 1993*) and both sympathetic and vagal influences (*Laborde, Mosley & Thayer, 2017*; *Malik et al., 1996*; *Shaffer & Ginsberg, 2017*). A comprehensive literature review conducted by *Reyes del Paso et al. (2013)* challenged this interpretation that the LF and LF/HF ratios reflect sympathetic activity and autonomic balance, respectively, and suggested that the LF component of the HRV is mainly determined by the parasympathetic system.

## Effortless or effortful attention

According to *Ullén et al. (2010)*, the co-activation of the SNS and the PNS results from the interaction between positive affect and high attention, which leads to a state of effortless attention. The flow experience is characterized by heightened concentration and heightened attention. Specific patterns of activity, like an increased heart rate, decreased HRV, shallow respiration, and increased facial EMG activity, are signs of mental effort (*Aasman, Mulder & Mulder, 1987*; *Backs & Seljos, 1994*; *Veltman & Gaillard, 1998*; *Waterink & Van Boxtel, 1994*), which are different from the observed results in studies concerned with flow. In contrast to the idea of effortless attention, *Keller et al. (2011)* linked reduced HRV to increased mental effort during the experience of flow in a computerized knowledge task. During simulation-based training on the use of the *enterprise resource planning software* with the three levels of difficulty (under-challenged, flow, and over-challenged), *Léger et al. (2014)* reported less mental effort during flow. According to their results, participants who exhibited smaller variations in their EDA levels (i.e., being more emotionally stable), lower HR, and higher HRV (indicative of less mental effort) were reported to be more likely to be cognitively absorbed. *Peifer et al. (2014)* also reported a positive linear relationship between HRV values (HF-HRV) and flow in a computer task (Cabin Air Management System Simulation). The participants' stress levels were manipulated via the Trier Social Stress Test (TSST) before they performed the task. After the task, their flow experience was evaluated with the FKS questionnaire. Flow was associated with increased HF-HRV, reflecting a decrease in mental effort. These results contradicted the findings by Keller and colleagues (*2011*), who found a negative relationship between flow and parasympathetic activity. The different findings indicate that during a difficult level of a computerized knowledge task or game, participants may not perceive as much stress and threat as they might experience during the TSST, which is designed to create considerable social anxiety.

Keller's study involved a relatively small number of participants and operationalized flow through skill-challenge balance in the flow condition, without actually measuring self-reported flow experiences, which might affect the validity of their findings.

*Harris, Vine & Wilson (2016)* explored whether concentration during flow is related to objective indices of effortful attention processing in a simulated car-racing task with the three standard levels of difficulty (under-challenged, flow, and over-challenged). The FKS questionnaire was used to check for experimental manipulation. The authors reported significantly higher flow scores for the flow condition during the game. The observed higher mental effort (lower HRV) and more focused attention (more focused eye gaze) along with less self-reported subjective effort in the flow condition than in the over-challenged condition suggested that the experience of flow is based on an efficient, but effortful, engagement of attention. The link between attention and flow was also examined by *De Sampaio Barros et al. (2018)* to see whether flow mobilizes attentional resources while playing two video games, *Tetris* and *Pong*. The authors added an ''autonomy'' condition to the traditional under-challenged, flow, and over-challenged conditions. They argued that giving the opportunity to determine the difficulty level is an important factor for experiencing flow. However, the flow scores measured by the FKS questionnaire in the flow (pre-selected) and autonomy (self-selected) conditions were similar, albeit greater than in the under-challenged and over-challenged conditions. The HR significantly increased with task difficulty, and the HRV was lower during the autonomy level than during the other conditions for both games, suggesting higher mental effort during autonomy.

## The inverted U-shaped relationship between flow and the stress system

A number of studies on the physiology of flow found associations between flow and physiological activation of the stress system. Keller and colleagues (*2011*) reported that a state of flow while playing a game involves high levels of tension reflected by higher salivary cortisol levels (increased HPA-axis activation). In the second experiment, the authors utilized *Tetris* in the three aforementioned conditions to see whether high involvement during the flow experience was associated with increased salivary cortisol levels. Higher cortisol levels were reported for the flow and over-challenged conditions. By combining the stress model with the flow model, *Peifer et al. (2014)* suggested an inverted U-shaped curve between LF-HRV and cortisol level on one hand and the flow experience assessed by the FKS questionnaire on the other hand, revealing moderate LF-HRV and cortisol levels in flow and low and high LF-HRV and cortisol values during under-challenged and over-challenged conditions, respectively.

The functional association between HRV factors (LF-HRV and HF-HRV) and flow (measured by the FKS questionnaire) was also assessed during a driving-simulation game (*Tozman et al., 2015*). The task used was a driving simulator chosen from the sporting-race, video-game package *Rfactor* with the three fixed levels of difficulty. An increase in task difficulty caused a decrease in the HF-HRV and LF-HRV components. In contrast to the findings by *Peifer et al. (2014)*, which showed an inverted U-shaped relation between flow and HRV measures, there was a negative linear connection between LF-HRV and flow

when the conditions for flow were met (flow condition) and an inverted U-shaped relation between LF-HRV and HF-HRV, on the one hand, and flow, on the other hand, when demands exceeded the skill level (over-challenged condition) (*Tozman et al., 2015*). In a VR game, *Bian et al. (2016)* reported results similar to the previous studies showing that increased HR, HRV, and respiratory rate (RR), as well as shorter inter-beat intervals (IBI), predict an increase in the flow score as assessed by the FKS questionnaire. An inverted U-shaped function between LF-HRV and HF-HRV and flow was also reported, highlighting moderate LF and HF-HRV levels for high flow scores and both low and high values of LF and HF-HRV for low-flow scores. The authors stated that the physiological aspects of flow in VR games may be particularly affected by the VR environment (*Bian et al., 2016*).

## Neural correlates of flow states

There is still considerable conceptual ambiguity concerning the possible brain mechanisms involved in the flow experience. Here we are going to discuss the main hypotheses established in the literature. Given the effortlessness and automatic characteristics of flow, *Dietrich (2004)* argued that such an optimal performance state is controlled through an implicit rather than an explicit information-processing system in the brain. The explicit system, which is associated with higher-order cognitive functions, is rule based, can be verbalized, is connected to conscious awareness, and is supported by frontal-lobe activation. In contrast, the implicit system is skill based, cannot be verbalized, is inaccessible to conscious awareness, and is supported primarily by the basal ganglia. Dietrich proposed that inhibition of the explicit system and transient hypofrontality is a necessary prerequisite for the experience of flow (*Dietrich, 2004*). The synchronization theory of flow proposed by *Weber et al. (2009)* specifies neuropsychological processes of the flow experience, considering that it is characterized by intense concentration and autotelic activity. This theory is based on Posner's tripartite theory of attention involving executive, alerting, and orienting networks (*Posner et al., 1987*). Accordingly, the optimal and gratifying experience of flow results from synchronized activity in the attentional and reward networks under the balanced skill-challenge condition (*Weber et al., 2009*). *Csikszentmihalyi (1975)* described the flow experience as "self-forgetfulness" or "loss of self-consciousness," highlighting the fact that, when the demands of the activity require the allocation of all attentional resources, attention is directed away from the self. Loss of self-awareness, as one of the important components of flow, sheds light on another interesting line of research that investigated the default mode network (DMN) activity during the flow experience (*Sadlo, 2016*). The activity of the DMN has been linked to self-referential thinking, and, therefore, declines in task-focused and goal-directed actions (*Goldberg, Harel & Malach, 2006*; *Raichle et al., 2001*). During moments of flow, DMN activity is thought to decrease, highlighting less self-referential processing (*Peifer, 2012*; *Sadlo, 2016*). Table 2 presents articles exploring brain activation during the flow experience.

## Transient hypofrontality

The transient-hypofrontality hypothesis proposed by *Dietrich (2004)* was addressed by a few studies. Applying brain-imaging techniques in blocks of mental-arithmetic tasks

Khoshnoud et al. (2020), *PeerJ*, DOI 10.7717/peerj.10520

**Table 2** Studies on neural correlates of the flow state: brain imaging investigations.

| Ref. | N subjects | Age mean/ range | Sample | Experiment type | Design | Measure | Neural correlates of flow |
|---|---|---|---|---|---|---|---|
| *Klasen et al. (2008)* | 18 | – | – | FPS game: Counter Strike | 12 min gameplay | fMRI | Correlation between game pleasure, cerebro-thalamic motor, & visual network activity |
| *Klasen et al. (2012)* | 13(m) | 18-26 | Healthy students | FPS game: Tactical Ops Assault on Terror | 12 min gameplay | fMRI | Activation of somatosensory networks & motor areas during situations with enhanced probability of flow |
| *Ulrich et al. (2014)* | 27(m) | 23 | Healthy students | Mental arithmetic task | 184 s Under-challenged, Flow, Over-challenged | Perfusion MRI | Increased activity in the left IFG, left putamen, & posterior cortical regions as well as decrease in MPFC and AMY in the flow condition |
| *Yoshida et al. (2014)* | 20(10f) | 21–25 | Healthy students | Tetris game | 4 min Under-challenged, Flow | fNIRS | Higher activation of VLPFC in the flow condition |
| *Harmat et al. (2015)* | 77(40f) | 27 | Healthy subjects | Tetris game | 6 min Under-challenged, Flow, Over-challenged | fNIRS, ECG, Res | No association between frontal cortical oxygenation & flow |
| *Ulrich, Keller & Grön (2016)* | 23(m) | 24 | Healthy students | Mental arithmetic task | 30 s Under-challenged, Flow, Over-challenged | fMRI , EDA | Increased activity in the left IFG, left putamen, & posterior cortical regions in the flow condition; decreased activity in MPFC, PCC and AMY in the flow condition |
| *De Sampaio Barros et al. (2018)* | 20 (7f) | 26 | Healthy adults | Tetris, Pong games | 3 min Under-challenged, Flow, Over-challenged, Autonomy | ECG, Res, NIRS | Higher activation in lateral PFC & deactivation in MPFC in the autonomy condition |

Khoshnoud et al. (2020), *PeerJ*, DOI 10.7717/peerj.10520

**Table 2** (*continued*)

| Ref. | N subjects | Age mean/ range | Sample | Experiment type | Design | Measure | Neural correlates of flow |
|---|---|---|---|---|---|---|---|
| *Ulrich et al. (2018)* | 22(m) | 24.9 | Healthy students | Mental arithmetic task | 170 s Under-challenged, Flow, Over-challenged | Perfusion MRI, tDCS | Increase in the flow index of lower-flow group under anodal midfrontal tDCS stimulation & stronger de-activation of AMY |
| *Gold & Ciorciari (2019)* | 11(m) 21(11f) | 29–31 | Trained & untrained gamers | FPS games: Counter Strike Global Offensive , Battlefield 4, Tetris game | 20 min & 3 min Under-challenged, Flow, Over-challenged | tDCS | Higher level of flow after the active tDCS over DLPFC & right parietal cortex |
| *Ju & Wallraven (2019)* | 31(m) | 24.8 | Healthy students | Car driving game | 3 min gameplay | fMRI | Positive correlations between the flow experience and brain activity in regions related to visual and spatial execution as well as attentional processes & negative correlations with the DMN's activity |

**Notes.**

fMRI, functional magnetic resonance imaging; MRI, magnetic resonance imaging; fNIRS, functional near-infrared spectroscopy; NIRS, near-infrared spectroscopy; ECG, electrocardiography; EDA, electrodermal activity; tDCS, transcranial direct-current stimulation; FPS, first person shooter; Res, respiration; IFG, inferior frontal gyrus; PFC, prefrontal cortex; MPFC, medial prefrontal cortex; AMY, amygdala; VLPFC, ventrolateral prefrontal cortex; DLPFC, dorsolateral prefrontal cortex; PCC, posterior cingulate cortex; DMN, default mode network; m, male; f, female.

with the aforementioned three levels of difficulty, *Ulrich et al. (2014)*; *Ulrich, Keller & Grön (2016)* reported a relative decrease in the activity of the medial prefrontal cortex (MPFC) in the flow condition. Other studies failed to confirm this finding. In a functional near-infrared spectroscopy (fNIRS) study, *Yoshida et al. (2014)* explored the activity of the prefrontal cortex (PFC) during flow and under-challenged levels of playing *Tetris* and failed to verify the transient hypofrontality hypothesis. Flow scores assessed with the Flow State Scale for Occupational Tasks (*Yoshida et al., 2013*) were higher in the flow than in the under-challenged condition. Significantly higher activation of the left and right ventrolateral prefrontal cortex (VLPFC) was reported during the final 30 s of flow than throughout the entire flow condition; the same trend was not observed during the under-challenged condition (*Yoshida et al., 2014*). *Harmat et al. (2015)* also failed to show an association between decreased activity in frontal-brain regions and flow while playing *Tetris*. None of their fNIRS analyses revealed associations between lower frontal-cortical activation and flow, suggesting that the neural substrates of flow may vary depending on the task (*Harmat et al., 2015*). *De Sampaio Barros et al. (2018)* recorded the cerebral hemodynamics of 20 volunteers while they played *Tetris* and *Pong*. The flow and autonomy playing conditions not only led to higher activation in the lateral PFC, but also to higher deactivation in the MPFC compared to the other conditions.

It seems that the neural signature of transient hypofrontality during flow is task dependent. In tasks which require sustained attention, a deactivation of prefrontal areas seems unlikely. *Gold & Ciorciari (2019)* investigated whether decreased excitability over the left dorsolateral prefrontal cortex (DLPFC) and increased excitability in the right parietal cortex during gameplay promotes an increased experience of flow as measured by the FSS questionnaire. Transcranial, direct-current stimulation (tDCS), which is a non-invasive, electrical-stimulation technique that modulates the activation of the cortical neurons under a probe electrode, was used to alter the excitability of the cortex. In the first experiment, they recruited trained gamers to play one of two FPS video games (*Counter Strike: Global Offensive* or *Battlefield 4*) in two sessions using active and sham tDCS stimulation. The second experiment was conducted with untrained gamers playing *Tetris* in under-challenged, flow, and over-challenged conditions. Both trained FPS and untrained *Tetris* players experienced significantly higher levels of flow after the active stimulation compared to the sham condition. The authors argued that inhibiting the DLPFC and the disruption of explicit executive functions resulted in improved implicit information processing and a more intense flow experience (*Gold & Ciorciari, 2019*).

## Synchronization of attentional and reward networks

One of the first studies to assess the neural correlates of enjoyment while playing video games by means of functional-magnetic-resonance imaging (fMRI) was conducted by Klasen and colleagues in 2008. The participants' brain activation was measured in relation to their subjective experience, which was assessed by having participants think aloud while they watched a replay of their gameplay session with an FPS game (*Counter-Strike: Source*). Reported game pleasure was correlated with cerebro-thalamic motor-network and visual-network activity (*Klasen et al., 2008*). In a subsequent study, *Klasen et al. (2012)*

focused on game events that contribute to the flow factors described by Csikszentmihalyi, and corresponding fMRI data were analyzed while participants played an FPS video game called *Tactical Ops: Assault on Terror*. Somatosensory networks and motor areas were jointly activated during flow-contributing events (*Klasen et al., 2012*). The authors interpreted that this sensorimotor activation reflects the stimulation of physical activity, suggesting deep involvement and immersion in the game. The activation patterns of individual flow factors included the reward system (putamen, caudate nucleus, and thalamus), error monitoring (anterior cingulate cortex; ACC), the orbito-frontal cortex (OFC), temporal poles (TP), and the motor system. Specifically, reward-system activation was detected during game events with a skill-challenge balance, i.e., during moments when the player was able to master the challenges of the game and had a rewarding experience. The involvement of the reward system along with motor areas in both studies was considered in line with the synchronization theory of *Weber et al. (2009)*. However, flow is a highly subjective phenomenon, and the second study did not examine the actual subjective flow experience, but events with an enhanced probability of flow.

*Ulrich, Keller & Grön (2016)* and *Ulrich et al. (2014)* also found increased activity in the inferior frontal gyrus (IFG, an executive attention structure) along with the left putamen (a region involved in reward processing), the anterior insula, and posterior cortical regions in the flow condition during the mental-arithmetic task. *Yoshida et al. (2014)* observed a higher activation of the right and left VLPFC during the flow condition while playing *Tetris*, which relates to reward and emotion processing in a state of flow. Considering the involvement of VLPFC in top-down attention (*Raz & Buhle, 2006*), one can interpret this as a co-activation of the attentional and reward networks during the flow experience (*Weber, Huskey & Craighead, 2016*). The results of the study by *De Sampaio Barros et al. (2018)* showed a significant positive correlation between the self-reported measure of attention and the average neural activation in the frontoparietal regions. Higher activation in the lateral PFC was reported in the flow and autonomy conditions while playing *Tetris* and *Pong* compared to the other conditions. In a custom-designed car game, *Ju & Wallraven (2019)* assessed the neural correlates of the flow experience with the flow subscale of the GEQ. Besides a baseline driving condition with a fixed structure, they designed three extra conditions to modulate the level of difficulty with one parameter (speed, obstacle, or tokens). Although no significant differences in the flow-subscale ratings were reported across conditions, the results of the fMRI analysis showed positive correlations between the flow scores and brain activity in regions related to visual (dorsal and ventral visual pathways) and spatial execution (middle and superior temporal gyri), as well as attentional processes (IFG, inferior and superior parietal lobules).

## Self-referential processing

Relating existing theories of the default mode network to the feeling of selflessness during flow, *Peifer (2012)* argued that the down regulation of task-irrelevant processes during the experience of flow should lead to decreased activity in these resting-state networks of the brain. First empirical evidence came from a magnetic-resonance-based, perfusion-imaging study by *Ulrich et al. (2014)*, who found that a relative decrease in activity in the MPFC (an

important structure for self-referential processing) and the amygdala (AMY) accompany the experience of flow in a mental-arithmetic task. The MPFC, the inferior parietal lobe, the posterior cingulate cortex (PCC), and the precuneus constitute the DMN (*Raichle et al., 2001*). A flow index that was specifically computed to represent the individually experienced level of flow correlated negatively with activity in the MPFC (indicating less self-related processing) and the AMY. The higher the subjective experience of flow, the greater the decrease in neural activity in the MPFC and AMY. The authors later explored the neural effects of flow experience at higher levels of temporal resolution using an fMRI block design with blocks of activation as short as 30 s (*Ulrich, Keller & Grön, 2016*). This study yielded similar results as their previous study, with the addition of decreased activation in the PCC, which altogether were interpreted as deep concentration and less self-referential processing along with less emotional arousal (reflected by down-modulation of AMY) during the flow experience. In the flow and autonomy playing conditions of the study by *De Sampaio Barros et al. (2018)*, decreased activity in the MPFC was also reported, which highlighted less self-referential processing during the experience of flow. *Ju & Wallraven (2019)* found negative correlations between the flow scores and brain activity in regions associated with the DMN in a car-driving game. Authors argued that the DMN as a task-negative network became more deactivated as players became more engaged in the game. Positive correlations between flow and activity in the insula in this study also indicated less self-awareness during moments of flow (*Ju & Wallraven, 2019*).

Ulrich and colleagues further explored the role of the MPFC in mediating flow experience using tDCS stimulation to interfere with the level of MPFC activation by modulating cortical excitability (*Ulrich et al., 2018*). During the above-mentioned mental-arithmetic task, current stimulation was applied over the frontal-central (Fpz) scalp position with three types of modulation: anodal (increase neuronal excitability), cathodal (decrease neuronal excitability), and sham (baseline). Flow experience was assessed along with the implementation of MR-based perfusion imaging while participants performed the task at three difficulty levels (under-challenged, flow, and over-challenged). There was no significant difference among stimulation types (sham, anodal, and cathodal tDCS) and the measured flow index across all subjects. After splitting the subjects into two groups based on the flow index in the sham condition (lower flow and higher flow), a significant increase in the flow index was reported in the lower-flow group under anodal tDCS stimulation. Anodal tDCS elicited a significantly stronger deactivation of the right AMY in this group compared to the higher-flow group (*Ulrich et al., 2018*).

### Neural oscillations and flow

*Nacke & Lindley (2010)* proposed an affective ludology context referring to investigations of affective player-game interaction. To address this issue, some studies have explored how electroencephalogram (EEG, assessment of cortical activity of the brain through electrodes placed on the scalp) signals can differentiate emotions from cognitive activity during gameplay. Specific neural oscillations in four frequency bands of EEG signal (delta, theta, alpha, and beta) were investigated as underlying neurophysiological mechanisms of the flow experience (see Table 3). These studies were mostly explorative without specific background

**Table 3** Studies on neural correlates of the flow state: neural oscillation investigations.

| Ref. | N subjects | Age mean/ range | Sample | Experiment type | Design | Measure | Neural correlates of flow |
|---|---|---|---|---|---|---|---|
| *Kramer (2007)* | 10 (5f) | 18–24 | Healthy students | Driving game | Playing trials | EDA, EEG | Greater left temporal alpha predicted performance level |
| *Chanel et al. (2011)* | 20(7f) | 27 | Healthy subjects | Tetris game | 5 min Under-challenged, Flow, Over-challenged | EDA, BP, Res, T, EEG | Distinct theta & beta power between conditions |
| *Nacke, Stellmach & Lindley (2011)* | 25(m) | 19–38 | Healthy students | FPS game: Half-Life 2 | 10 min Under-challenged, Immersion, Flow | EEG | Higher theta & delta power in the immersion condition |
| *Berta et al. (2013)* | 22 (5f) | 26.3 | Healthy students | Plane battle game | 4 min Under-challenged, Flow, Over-challenged | EEG | Lowest mean alpha & low-beta in the flow condition |
| *Wang & Hsu (2014)* | 20(10f) | 19–27 | Healthy students | Computer-based instruction | 7-9 min Under-challenged, Flow, Over-challenged | EEG | EEG attention value was correlated with overall flow and flow dimensions |
| *De Kock (2014)* | 20(m) | 16–45 | Healthy subjects | Car racing game: Need for Speed – Carbon | Low flow-performance, High flow-performance | EEG | Increased low-beta power in the sensori-motor; low-beta synchronization between all cortical connections for high-flow group |
| *Léger et al. (2014)* | 36 | – | Healthy students | Enterprise Resource Planning software | Under-challenged, Flow, Over-challenged | ECG, EDA, EEG | Higher alpha & lower beta in the flow condition |
| *Wolf et al. (2015)* | 35 (9f) | <36 | Table-tennis players | Motor imagery paradigm | 7 s video clips | EEG | Positive correlation between T4-T3 alpha asymmetry & flow score in the experts |
| *Metin et al. (2017)* | 20(7f) | 20–35 | Healthy subjects | Ping-pong game | 2 min Under-challenged, Flow | EEG | Greater theta & delta power in the flow condition |

Khoshnoud et al. (2020), *PeerJ*, DOI 10.7717/peerj.10520

Peer J

**Table 3** (*continued*)

| Ref. | N subjects | Age mean/ range | Sample | Experiment type | Design | Measure | Neural correlates of flow |
|---|---|---|---|---|---|---|---|
| *Katahira et al. (2018)* | 16(6f) | 21.9 | Healthy students | Mental arithmetic task | 184 s Under-challenged, Flow, Over-challenged | EEG | Increased theta activity in the frontal areas, moderate alpha activities in the frontal & central areas in the flow condition |
| *Moreno et al. (2020)* | 1 | 27 | Expert gamer | Portal game | 45 min gameplay | EDA , EEG | Increased beta activity during moments of goal attainment |

**Notes.**

ECG, electrocardiography; EEG, electroencephalography; EDA, electrodermal activity; Res, respiration; BP, blood pressure; T, temperature; SD, standard deviation; m, male; f, female.

theories. Some studies examined whether verbal-analytic processing is reduced during flow in accordance with the notion of peak performance and automaticity characteristics of the flow experience (specifically in athletes' motor responses) (*Harris, Vine & Wilson, 2017*; *Kramer, 2007*; *Wolf et al., 2015*). Temporal alpha asymmetry has been shown to relate to peak performance, especially in athletes (*Kerick, Douglass & Hatfield, 2004*). According to *Vernon, (2005)*, higher left-temporal-cortex alpha activity, which reflects decreased cortical activity in this region, is associated with improved performance, as it represents a decrease in internal verbalizations and increased visual-spatial processing in the right hemisphere. Among frequency bands, frontal theta activity (specifically frontal mid-line theta) was of particular interest. Frontal mid-line theta has been linked to cognitive control and concentration (*Brandmeyer, Delorme & Wahbeh, 2019*; *Cavanagh & Frank, 2014*) and may increase during the flow experience.

### Reduced verbal-analytic processing

*Kramer (2007)* studied neural correlates of peak performance (as associated with the state of flow) by exploring the power information of EEG signals in a car-driving game. A decrease in alpha power in the right-temporal lobe prior to a game trial predicted better game performance as reflected by an increase in visuo-spatial processing. Greater mean left-temporal alpha power ten seconds before a game trial resulted in improved performance. Once again, the players' subjective flow experience was not directly evaluated, but the high-performance intervals functioned as a proxy for the subjective states. *Wolf et al. (2015)* later linked states of highly-focused attention in athletes (one key component of the flow experience) to a reduced influence of verbal-analytical processes reflected by stronger relative left-temporal-cortex alpha power. In this study, 35 expert and amateur table-tennis players were asked to watch a 7-second video clip of a table-tennis player serving a ball and to imagine themselves reacting to it. A significant change towards lower T4–T3 alpha power (stronger right-temporal-cortical activity) at the beginning of the movement phase was reported in experts. This result, along with a positive correlation between T4–T3 alpha asymmetry and the flow score (measured by the FKS questionnaire) in the experts, was interpreted to reflect lower verbal analytic processing as associated with a higher degree of flow in expert table-tennis players.

### Delta and Theta frequency bands

*Chanel et al. (2011)* tried to classify the three states (under-challenged, flow, and over-challenged) induced by playing *Tetris* at three different challenge-skill levels. Although their EEG results indicated distinct theta power among conditions in some electrodes, no precise relationship to the individual states (under-challenge, flow, and over-challenge) was found. Nacke and colleagues probed the impact of different difficulty levels of a game on brainwave activity in an exploratory EEG study (*Nacke, Stellmach & Lindley, 2011*). The authors did not employ the difficulty modulation to create different levels of gameplay. Three gameplay conditions (under-challenged, immersion, and flow) of the game *Half-Life 2* were created based on specific level-design guidelines (LDGs). Theta and delta power were significantly higher in immersion than in flow and the under-challenged condition. Since the immersion condition of the game required navigating through landmarks, the

authors argued that high theta activity in this level might be attributed to its architectural complexity. In another study, EEG correlates of the flow state induced by playing a ping-pong video game were investigated at two levels, slow as under-challenge and fast as flow inducing (*Metin et al., 2017*). EEG-frequency power evaluations revealed higher mean theta power during the flow condition for all regions of interest and higher mean delta power in frontal, central, and parietal regions compared to the non-flow condition. The regional theta- and delta-frequency bands correlated positively with the absorption, enjoyment, and intrinsic-motivation subscales of the Turkish version of the FKS flow questionnaire. Regarding the two playing levels, a higher theta band was to be expected in the more difficult (flow) condition, as theta activity has been linked to concentration, working memory, and sustained attention, which increase with higher difficulty levels. *Katahira et al. (2018)* characterized the flow state by increased theta activity in the frontal areas. Employing a mental-arithmetic task used in the previous study (*Ulrich et al., 2014*) with three levels of difficulty, theta activity in the frontal areas was reported to be higher during the flow and the over-challenged conditions of the task compared to the under-challenged condition.

### Alpha frequency band

Alpha-power attenuation in the flow condition was seen as an indicator that the subject had entered into a flow state (*Berta et al., 2013*). Using a four-electrode EEG (F7, F8, T5, and T6), distinct states induced by a specifically designed plane-battle video game were analyzed with appropriate levels for the under-challenged, flow, and over-challenged conditions. The main differences among the three conditions were reported in alpha and low-beta frequency-band powers with the lowest alpha and low-beta in the flow state. There was no information regarding the region of observed distinct frequency powers among conditions. Self-assessed flow scores of the GEQ revealed significant differences at under-challenged and over-challenged levels, but failed to distinguish the flow level. *Léger et al. (2014)* explored the relationship between EEG and flow in a simulation-based training session at the three levels of difficulty. Subjects with high-alpha and low-beta activity reported higher cognitive-absorption scores. The authors argued that these results demonstrated a more relaxed and less vigilant state in the learners.

### Beta frequency band

*Wang & Hsu (2014)* explored the state of flow during a computer-based instruction paradigm utilizing EEG to see whether the attention score captured by the EEG signal was associated with the flow score assessed by the virtual-course flow measure. Participants completed three lessons on computer-based Excel instructions with under-challenged, flow, and over-challenged contents. The EEG attention value derived from the beta-wave-brain activity at the Fp1 electrode correlated with the flow dimensions of enjoyment, focused attention, involvement, and time distortion. However, the correlation coefficient was small, and the authors argued that the attention value did not precisely represent the flow experience and comprised only one component. *Léger et al. (2014)* found low-beta activity associated with a higher flow score, highlighting a less vigilant state in the learners. Utilizing an adapted version of the WOLF questionnaire, *De Kock (2014)* evaluated the flow
experience of participants playing a continuous visuomotor computer game (*Need for Speed Carbon*). EEG-signal activity at prefrontal, sensorimotor, parietal, and occipital regions was compared between low-flow-/low-performance and high-flow- /high-performance groups. The high-flow condition was associated with increased low-beta power in the sensorimotor cortex, as well as low-beta synchronization among all cortical connections. The shift in low-beta power in the sensorimotor area was connected to fluent and coordinated motion. Synchronized low-beta connections in all cortical regions in the high-flow condition indicated optimized transmission of neural information throughout the brain, ensuring smooth, accurate, and effortless motor execution (*De Kock, 2014*). Increased beta activity during flow-like states was also reported in a single-case study by *Moreno et al. (2020)* highlighting higher cognitive engagement during moments of flow.

## Implementation of dual-task paradigms

Although the above-mentioned studies have provided neural signatures of the flow state, they all face a similar limitation. Their methodology cannot discern between internal flow and the external task conditions that facilitate the experience of flow. A skill-challenge balance is considered a prerequisite to inducing flow, but it does not guarantee that an individual will enter the flow state. It has been demonstrated that different factors, but especially the methodology, can affect the association between skill-challenge balance and flow (*Fong, Zaleski & Leach, 2015*). In some studies in which difficulty modulation was used for creating under-challenged, flow, and over-challenged conditions, the adaptive playing condition was considered as flow inducing without any post-manipulation check to see whether participants really experienced flow in their experimental set-up (e.g., *Chanel et al., 2011*). This fact led to the application of techniques that indirectly measure the extent to which subjects experience flow by assessing their levels of attentional focus. Based on the flow theory (*Csikszentmihalyi & Csikszentmihalyi, 1992*), focused attention during the experience of flow leads to complete absorption in an activity to the extent that one does not allocate attentional resources to irrelevant external stimuli. During under-challenged or over-challenged states, attentional disengagement from the task makes it more likely that an individual will pay attention to irrelevant stimuli. These considerations led to an interesting line of research using dual-task paradigms to indirectly measure electrophysiological correlates of the flow experience (see Table 4). Secondary-task reaction times were suggested as reliable and valid measures of available attentional resources (*Weber et al., 2018*).

Castellar and colleageus utilized an auditory oddball paradigm as a secondary task to investigate attention while subjects played a game as a primary task (*Castellar et al., 2016*). Participants were requested to play the game *Star Reaction* in under-challenged, flow, and over-challenged conditions while simultaneously responding to a rare sound in the auditory oddball task. The greater the absorption in the primary task, the slower the reaction times and more errors registered in the detection of oddball sounds. Event-related-potential (ERP) analysis showed that the maximal frontocentral negative deflection after the response onset was significantly delayed during the flow condition compared to the other two conditions in the correct-responses trials, reflecting delayed attention

**Table 4  Studies on neural correlates of the flow state: dual-task paradigms.**

| Ref. | N subjects | Age mean/ range | Sample | Experiment type | Design | Measure | Neural correlates of flow |
|------|-----------|-----------------|--------|-----------------|--------|---------|---------------------------|
| *Castellar et al. (2016)* | 18(9f) | 28.5 | Healthy subjects | Star reaction game & Auditory oddball detection | Under-challenged, Flow, Over-challenged | EEG | Delayed maximal frontocentral negative deflection after the response onset in the flow condition |
| *Yun et al. (2017)* | 29(5f) | 23.5 | Healthy subjects | FPS game: Call of Duty & Random beeping sound | 30 min Low challenge/ High challenge | EEG | Suppressed evoked potential during self-reported experience of flow |
| *Bombeke et al. (2018)* | 18(3f) | 25 | Healthy students | FPS game Counter-Strike: Global offensive & Auditory oddball detection | 8 min Under-challenged, Flow, Over-challenged | EEG | Mid-line P300 amplitude smaller in VR compared to playing in 2D in the flow condition |
| *Huskey et al. (2018)* | 18(m) | – | Healthy students | Asteroid Impact game & Secondary reaction time | 2 min Under-challenged, Flow, Over-challenged | fMRI | Higher activity in DLPFC, SPL, DAI, & putamen in the flow condition |

**Notes.**

EEG, electroencephalography; fMRI, functional magnetic resonance imaging; ERSP, event-related spectral perturbation; VR, virtual reality; DLPFC, dorsolateral prefrontal cortex; SPL, superior parietal lobe; DAI, dorsal anterior insula; m, male; f, female.

reallocation to the primary task during flow. Significant increases in the midfrontal alpha power during the flow condition may well indicate the intrinsic rewarding nature of the flow experience (*Castellar et al., 2016*). A study by *Yun et al. (2017)* extended the secondary-task idea by adding a passive random beeping sound while subjects played an FPS game (*Call of Duty: Modern Warfare 2*). Complete absorption in the game world was expected to lead to the neglect of the game-irrelevant sensory stimulation from the real world, which is reflected by the suppression of auditory evoked potentials (AEPs) of EEG signals elicited by random beeps. Due to the insufficient number of trials and background noise, typically detected AEPs were not observable, and the authors instead analyzed event-related spectral-perturbation (ERSP) suppression at low frequencies in the flow trials. A significant correlation was reported between the suppressed evoked potentials derived from ERSP and the self-reported experience of flow. By utilizing source-localization algorithms, the activation of the ACC and the temporal pole was reported during flow trials only in the beta-frequency range. Subjective flow ratings also positively correlated with activation in these regions, suggesting a link between the flow experience and high concentration, focused attention, and less self-referential processing (*Yun et al., 2017*).

Auditory oddball sounds were also applied as a secondary task in a VR gaming context to explore attentional allocation during the experience of flow (*Bombeke et al., 2018*). Participants played the shooter game *Counter-Strike: Global Offensive* under three conditions (under-challenged, flow, and over-challenged) both in a 2D and a VR set-up while they were simultaneously asked to respond to the oddball sounds. Their results did not replicate the outcome of the previous study by *Castellar et al. (2016)*, as they reported slower reaction times and more errors in the flow condition. A marginally significant posterior mid-line P300 amplitude was recorded during VR compared to playing in 2D during the flow condition. The flow ratings measured with the FQ scale did not show any significant differences among the different gaming conditions, and it is unclear whether participants really experienced under-challenge, flow, and over-challenge in this set-up. *Huskey et al. (2018)* recorded the greatest intrinsic reward (measured by the Autotelic Experience Subscale of the FSS questionnaire) and longer reaction times during the flow condition by applying a secondary-task reaction-time (STRT) procedure while subjects played the game *Asteroid Impact* at the three levels of difficulty. The flow condition elicited significantly greater activity in the areas related to cognitive control (DLPFC), orienting attention (superior parietal lobe; SPL), attentional alerting (dorsal anterior insula, dAI), and reward networks (putamen). In line with the synchronization theory of flow, the low-difficulty condition evoked activity in DMN structures which was absent in the high-difficulty condition (*Huskey et al., 2018*).

## DISCUSSION

We conducted a comprehensive review of the current literature on the underlying electrophysiological and neural mechanisms of the experience of flow. Although a number of physiological and neural measures could potentially be considered as markers of flow, it is difficult to relate them to a unified mechanism underlying this mental state. Flow is

a complex state that requires the involvement of distinct cognitive subfunctions, which in turn necessitates the activation of different physiological and neural systems. Here we categorized some of these distinct physiological and cognitive subfunctions which were addressed by most of the studies.

## The state of positive valence and heightened arousal

Activity in the smiling (ZM, positive association) and frowning facial muscles (CS, negative association) and greater respiratory depth during flow states represent positive affect (*De Manzano et al., 2010*; *Harmat et al., 2015*; *Kivikangas, 2006*; *Mauri et al., 2011*; *Nacke & Lindley, 2008*). The pattern of arousal modulation, however, is complex and varies according to how studies used it to distinguish flow states from taxing experiences, such as stress. Peifer and colleagues (*2014*) proposed an inverted U-shaped function between the flow experience and physiological arousal. If we consider the relationship between flow and performance (*Csikszentmihalyi, Abuhamdeh & Nakamura, 2005*; *Engeser & Rheinberg, 2008*; *Jin, 2012*; *Keller & Bless, 2008*; *Landhäußer & Keller, 2012*), this pattern aligns well with the Yerkes-Dodson Law, which proposes an inverted U-shaped association between arousal and performance (*Yerkes & Dodson, 1908*). Nevertheless, findings concerning the sympathetic and parasympathetic reflections of arousal are heterogeneous, given that both linear (*Chanel et al., 2011*; *De Manzano et al., 2010*; *De Sampaio Barros et al., 2018*; *Keller et al., 2011*; *Tian et al., 2017*) and inverted U-shaped (*Bian et al., 2016*; *Peifer et al., 2014*; *Tozman et al., 2015*) associations have been reported. On the other hand, EDA—a robust indicator of sympathetic arousal (*Critchley & Nagai, 2013*)—has been shown to positively correlate with flow, reflecting heightened sympathetic arousal during moments of flow (*Léger et al., 2014*; *Moreno et al., 2020*; *Nacke & Lindley, 2010*; *Ulrich, Keller & Grön, 2016*).

One possible explanation of the linear function between arousal and the flow experience is that playing a game in a laboratory setting, even at a higher level of difficulty, might not be perceived as a threat and thus fails to elicit high levels of arousal at high levels of difficulty. Based on the biopsychosocial model of challenge and threat, *Tozman & Peifer (2016)* suggested using framing techniques to manipulate challenge appraisal in a game and create a threatening situation during gameplay. The question is how a framing context affects flow experience, given that external impositions, such as threat, negative feedback, and an imposed deadline, might negatively influence intrinsic motivation and, consequently, the flow experience (*Di Domenico & Ryan, 2017*). Studies investigating the relationship between flow and salivary cortisol levels (*Keller et al., 2011*; *Peifer et al., 2015*; *Peifer et al., 2014*) also suggest that this relationship is moderated by the type of intervention, personal characteristics, and the interaction of both (*Brom et al., 2014*). It is also possible that the internal motivation for gameplay was adversely affected by the experimental setup, as players participated in the experiment not for the pleasure of the game, but for the specific context. Being in an artificial experimental situation and receiving external rewards (e.g., monetary compensation) may suppress arousal. It is worth mentioning that the LF-HRV which was considered a marker of sympathetic activity in the study by Peifer and colleagues (*2014*), was determined mainly by activity in the parasympathetic system (*Reyes del Paso et al., 2013*). Future studies should consider more robust indicators of sympathetic arousal to

evaluate the relationship between flow and physiological arousal. The pre-ejection period of cardiovascular activity was suggested as a reliable indicator to clarify this inconsistency (*Tozman & Peifer, 2016*). Generally speaking, the simultaneous presence of heightened arousal and positive valence can distinguish flow from the experiences of under-challenged and over-challenged conditions.

## The joyous state of focused attention

Flow as a state of complete concentration during a balanced skill-challenge condition necessitates a high degree of attention that is understood to be effortless. Both flow and mental effort increase with increasing task difficulty (*Tozman & Peifer, 2016*), but the specific pattern of activity in the autonomic nervous system observed during the flow experience (e.g., decreased heart period with deep respiration) is different from the pattern associated with mental effort (e.g., decreased heart period, lower HRV, and rapid and shallow respiration). Studies that found lower HRV in the flow condition explained this phenomenon in the light of greater mental effort during the experience of flow (*De Sampaio Barros et al., 2018*; *Harris, Vine & Wilson, 2016*; *Keller et al., 2011*). In contrast, studies that observed a higher HRV suggested lower mental effort (*Bian et al., 2016*; *Peifer et al., 2014*). The inconsistent findings regarding mental effort and flow experience could be partly traced back to the imprecise measures used for assessing mental effort. Although HRV has been found to be a sensitive indication of mental effort (*Aasman, Mulder & Mulder, 1987*; *Backs & Seljos, 1994*; *Waterink & Van Boxtel, 1994*). *Veltman & Gaillard, (1998)* argued the opposite, as it can be affected by respiratory activity. For instance, during moments with more rapid respiration, differences in mental effort measured by HRV might be overestimated. It is, therefore, necessary to test more precise measures of the suppression of the cardiovascular control system resulting from mental effort. Blood glucose and pupil dilation were suggested as sensitive measures to explore mental effort (*Saproo et al., 2016*; *Tozman & Peifer, 2016*) which have not been investigated in the context of flow.

*Harris, Vine & Wilson (2016)* demonstrated that subjective and objective attentional effort might separate from one other. Focused eye gaze (increased attention) and lower HRV (higher mental effort) reported during the flow condition did not match the lower effort scores obtained through self-report (*Harris, Vine & Wilson, 2016*). *Ullén et al. (2010)* suggested that this may occur as a result of an interaction between positive valence and focused attention. In a state of positive affect, a task requiring considerable mental attention might be experienced as less effortful than when accompanied by a state of negative affect. Observed co-activation of the sympathetic (reflected by decreased HP) and parasympathetic systems (reflected by deep respiration) aligns well with this suggestion. Considering the role of the PFC in attention and concentration, the experience of flow was found to be associated with increased activity in this integrative frontal area of the cortex (*Klasen et al., 2012*; *Ulrich, Keller & Grön, 2016*; *Ulrich et al., 2014*; *Yoshida et al., 2014*). Frontal-midline-theta activation has been linked to concentration, working memory, and sustained attention (*Cavanagh & Frank, 2014*). High theta activity reported during flow (*Katahira et al., 2018*; *Metin et al., 2017*; *Nacke, Stellmach & Lindley, 2011*) may well reflect focused attention. We consider the flow state to be accompanied by an efficient attentional

effort and that the coupled activity of the sympathetic and parasympathetic nervous systems can be used to distinguish this joyous state of focused attention from a purely onerous mental experience.

## Synchronized activation of attentional and reward networks

Flow is considered to be a state of focused attention which is intrinsically rewarding, as the flow-inducing task is performed for its own sake. Some studies support the synchronization theory of flow (*Weber et al., 2009*) by showing the joint activation of frontoparietal attention networks (e.g., IFG and inferior parietal lobe) and reward networks (e.g., putamen, thalamus) during the flow experience (*Castellar et al., 2016*; *De Sampaio Barros et al., 2018*; *Huskey et al., 2018*; *Ju & Wallraven, 2019*; *Klasen et al., 2012*; *Ulrich, Keller & Grön, 2016*; *Ulrich et al., 2014*; *Yoshida et al., 2014*). A positive correlation between dopaminergic receptor availability in the striatum and putamen and flow proneness supports this theory and shows that the experience of flow is intrinsically rewarding (*De Manzano et al., 2013*).

## Automaticity

Inhibition of the explicit system and the transient hypofrontality theory (*Dietrich, 2004*) received partial empirical support from studies on the neural mechanisms underlying the experience of flow (*Gold & Ciorciari, 2019*; *Ulrich, Keller & Grön, 2016*; *Ulrich et al., 2014*). Other studies failed to find transient hypofrontality during the state of flow (*Harmat et al., 2015*; *Yoshida et al., 2014*). Fluent, smooth, and effortless motor performance was related to increased low-beta power in the sensorimotor cortex and low-beta synchronization among all cortical connections (*De Kock, 2014*). This hypothesis might be an oversimplification of the flow state or only be related to specific situations. During tasks with high demands on executive control, a decoupling of actions from conscious effort and controlled attention is unlikely to happen. It has been suggested that the decrease in frontal functions is more likely to occur when the action becomes more automatic (*Harris, Vine & Wilson, 2017*). This means that transient hypofrontality might happen after prolonged practice.

## Loss of self-awareness

A promising consistent outcome of neural research on flow experience is the deactivation of the DMN, specifically the MPFC, which indicates less self-referential processing during the flow experience (*De Sampaio Barros et al., 2018*; *Ju & Wallraven, 2019*; *Sadlo, 2016*; *Ulrich, Keller & Grön, 2016*; *Ulrich et al., 2014*; *Ulrich et al., 2018*). It has been stated that during the performance of cognitively demanding tasks, the activity of the central executive network and the salience network increases whereas DMN activity decreases (*Sridharan, Levitin & Menon, 2008*). Activity in the DMN is reported to be associated with a relaxed mind, mind-wandering, and self-referential thinking, which are reduced in task-focused and goal-directed actions (*Goldberg, Harel & Malach, 2006*; *Raichle et al., 2001*). Reduced activity was found in the DMN during focused sensory perception (*Goldberg, Harel & Malach, 2006*), which reflects the loss of self during the activity. Activation of DMN regions was also reported during a boredom-induction task, suggesting a relation between mind wandering and DMN activity (*Danckert & Merrifield, 2018*). Several studies discussed the role of the MPFC and its relatively decreased activity in self-referential processing

(*Goldberg, Harel & Malach, 2006*; *Gusnard et al., 2001*; *Raichle et al., 2001*). This is strongly related to *Csikszentmihalyi*'s (*1990*) dimension of loss of self-awareness in flow theory. High concentration and focused attention demanded by the task at hand restrict resource allocation for task-irrelevant demands like body and self-awareness. *Sridharan, Levitin & Menon (2008)* stated that the salience network, including the VLPFC and the anterior insula (AI), is involved in shifts between the DMN and cognitive executive networks acting as an outflow hub at the junction of both networks. This theory is further confirmed by the positive correlation between the flow experience and the increase in activity in the insular cortex, especially in the anterior insula (*Huskey et al., 2018*; *Ju & Wallraven, 2019*; *Ulrich, Keller & Grön, 2016*). Consequently, higher activity in the anterior insula might show disengagement of the task-irrelevant DMN regions during the experience of flow. Activity in anterior and posterior parts of the insula was linked to time perception (*Wittmann et al., 2010*), and the anterior regions were shown to associate with the experience of bodily self-awareness (*Craig, 2009*). A study by *Berkovich-Ohana et al. (2013)* reported that timelessness during meditation is accompanied by higher theta activation in the right insula. The role of the anterior insula in the experience of flow should be clarified by further investigations.

Another important issue here is that lower self-referential information processing is associated with decreased neural activity in the amygdala during flow (*Ulrich, Keller & Grön, 2016*; *Ulrich et al., 2014*; *Ulrich et al., 2018*). Given the amygdala's mediating role in the perception of emotions (*Morris et al., 1996*), reduced activity in AMY likely reflected decreased emotional arousal associated with the experience of flow. Lower self-awareness may reduce the threat response and increase positive emotions (*Sadlo, 2016*; *Ulrich, Keller & Grön, 2016*). Reduced awareness of the self is also reported to contribute to improved athletic performance (*Harris, Vine & Wilson, 2017*). The close relationship between flow experiences and performance (*Engeser & Rheinberg, 2008*; *Jin, 2012*; *Landhäußer & Keller, 2012*) suggests that reduced self-awareness and DMN activity is one of the underlying key features of the flow experience.

## Task dependency

Some of the inconsistencies in results can be explained by different experimental designs used in different research approaches. While some studies used continuous playing and correlational analysis, others preferred a difficulty-modulation approach in which they designed three game levels corresponding to the under-challenge, flow, and over-challenge categories of the flow model. The way the skill-challenge balance is operationalized in these studies directly influences flow. Studies frequently used global flow scales, like the FSS or the FKS questionnaires, to measure participants' levels of flow. A few studies merely applied some individual items or subscales of these surveys to assess the subjective experience (*Keller et al., 2011*; *Ulrich, Keller & Grön, 2016*; *Ulrich et al., 2014*), and some others did not employ any measure for the evaluation of flow and theorized that the skill-challenge-balance condition induces a flow state without further controlling for the effects of this manipulation (e.g., *Chanel et al., 2011*; *Nacke & Lindley, 2008*; *Nacke, Stellmach & Lindley, 2011*). In a meta-analysis, *Fong, Zaleski & Leach (2015)* reported that

the correlation between flow and optimal balance is higher when a global flow scale and one of its subscales of challenge-skill fit was used to operationalize a skill-challenge fit. The length of the experimental blocks, ranging from 30 s to 12 min, is another limitation which leads to strong variations in the strength of the flow experience. It has been stated that participants require a minimum of 25 min to get into the flow state (*Bisson, Tobin & Grondin, 2012*; *Tobin, Bisson & Grondin, 2010*; *Yun et al., 2017*). The next concern is that different paradigms or games require the involvement of different cognitive functions, which in turn affect the outcomes of physiological and neural activity. *Peifer (2012)* argued that, since the physiological and cognitive demands of the flow-inducing activities are different, the neurophysiology of "optimal functioning" between them differs. First-person shooters (FPSs), like *Half-Life 2* (HL2) or *Counter-Strike: Source* (CS:S), require more complex interactions than, for instance, *Tetris* or *Pong*. Virtually anyone could pick up *Tetris* or *Pong* and play them right away, since the player only needs to push a few buttons. In FPSs, players typically control the character with a combination of mouse and keyboard that takes practice to use. Moreover, there are differences between the contents of the games used in the above-mentioned studies that require players to use different cognitive functions while playing. FPSs are three-dimensional games in which navigation is crucial, since the player only sees a small portion of the total space at any given time, and challenges are often hidden from view until they are close to the player or in their line of sight. In contrast, *Tetris* and *Pong* are two-dimensional and belong to the category of single-screen games, since all the relevant information is displayed simultaneously on the screen.

## Remaining issues and future research considerations

It is important to note that most of the methodologies mentioned above cannot discriminate between internal states of flow and the external conditions that help induce a flow experience. Designing specific levels for the experiments (corresponding to under-challenged, flow, and over-challenged) that are directly related to the amount of skill-challenge balance does not guarantee that people will enter a flow state in the flow condition. The subjective experience of flow in the flow condition should be directly assessed using self-report flow scales in future studies to determine whether the participants were able to enter into a full flow state or not. One could consider adding objective measures other than neural or physiological markers to isolate the state of flow. One type of objective measure was designed by applying a secondary reaction-time task to assess the level of attentional focus during gameplay (*Bombeke et al., 2018*; *Castellar et al., 2016*; *Huskey et al., 2018*; *Yun et al., 2017*). Longer reaction times and more errors in the secondary task were reported to correlate with the subjective experience of flow (*Castellar et al., 2016*; *Huskey et al., 2018*).

The associations between performance and arousal (*Yerkes & Dodson, 1908*) and flow and arousal (*Peifer et al., 2014*) suggest a close relationship between the flow experience and performance. The direction of this association has not yet been specifically investigated. It has been demonstrated that flow is a state of high concentration and that a sense of control can actually motivate subjects to improve their performance (*Engeser & Rheinberg, 2008*; *Jin, 2012*; *Landhäußer & Keller, 2012*). While the association between flow and optimal

performance has been described in academic activities, music, and sports (*Landhäußer & Keller, 2012*), few studies have reported a relationship in the gaming context (*De Kock, 2014*; *Engeser & Rheinberg, 2008*; *Jin, 2012*; *Keller & Bless, 2008*; *Yun et al., 2017*). Some studies failed to find an association between flow and optimal performance (*Harris, Vine & Wilson, 2016*; *Katahira et al., 2018*; *Ulrich, Keller & Grön, 2016*; *Ulrich et al., 2014*), reporting medium levels of performance during the flow condition. A positive association was mostly reported in studies in which components of subjective flow were directly measured instead of assessing behavioral levels of challenge and skills (*De Kock, 2014*; *Yun et al., 2017*). The causal relationship between flow and performance cannot be tested in typical cross-sectional experimental paradigms using difficulty manipulations; a longitudinal design is required to assess causality (*Keller & Bless, 2008*; *Landhäußer & Keller, 2012*). Performance could also provide an objective measure that, in combination with other measures (e.g., physiological and neural indices, subjective self-reports, and secondary reaction times), could precisely capture the actual emergence of flow. We argue that future studies should consider using objective measures beside subjective scales and self-reports to capture the actual emergence of flow.

## CONCLUSIONS

This review provides an overview of physiological and neural findings during the flow experience and integrates the empirical results to explain the underlying mechanisms of this complex state. We separated distinct physiological and cognitive subfunctions involved in the experience of flow. We conclude that flow is a positive mental state characterized by heightened arousal, focused attention, synchronized activity in the brain's attention and reward networks and results in automatic action control with less self-referential processing. Combining objective measures with retrospective questionnaires seems essential to capture the actual emergence of flow. The important role of focused attention during moments of flow necessitates employing dual-task paradigms to distinguish internal flow phenomena from external situations inducing flow.

## ACKNOWLEDGEMENTS

We thank one anonymous reviewer as well as Dr. Birte Thissen for the constructive and detailed criticism of our manuscript that helped shaping the final version.

### Funding

This study was funded by the EU, Horizon 2020 Framework Program, FET Proactive (VIRTUALTIMES consortium, grant agreement Id: 824128 to Marc Wittmann). The funders had no role in study design, data collection and analysis, decision to publish, or preparation of the manuscript.

## Grant Disclosures

The following grant information was disclosed by the authors:

EU, Horizon 2020 Framework Program, FET Proactive (VIRTUALTIMES consortium: 824128.

## Competing Interests

The authors declare there are no competing interests.

## Author Contributions

- Shiva Khoshnoud conceived and designed the experiments, performed the experiments, analyzed the data, prepared figures and/or tables, authored or reviewed drafts of the paper, and approved the final draft.
- Federico Alvarez Igarzábal conceived and designed the experiments, authored or reviewed drafts of the paper, and approved the final draft.
- Marc Wittmann performed the experiments, authored or reviewed drafts of the paper, and approved the final draft.

## Data Availability

This is a literature review.

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
