# Peer review of "Peripheral-physiological and neural correlates of the flow experience while playing video games: a comprehensive review"

_PeerJ, doi:10.7717/peerj.10520_

## Round 0.1 · original submission · Major Revisions

I have been able to obtain reviews from two experts in the field of flow. Both were generally positive, believing that your manuscript represents a strong effort, and that such a review will be of general interest to the field of flow researchers. However, both also highlighted a number of concerns, which I share. Please prepare a revision of the paper, taking into account the comments and suggestions of the reviewers, and with specific attention to these issues:

a) overall organization of the manuscript: while the manuscript is already quite long, there is a lack of synthesis or summary for each section, with all integration occurring in the Discussion section. One potential option might be to integrate the literature review and discussion points together, which may even allow for a shortening of the manuscript. Both reviewers also highlighted major organizational concerns that need to be addressed to improve readability.

b) addition of critical interpretation: both reviewers highlight a lack of critical interpretation of individual studies but also of larger topics. Adding such interpretation will help the reader to make sense of the literature.

c) identification of unresolved questions and gaps that would help the field move forward.

With regard to the many specific suggestions that the reviewers have made, I would note that while you may choose to address some of the issues in a different manner, I would ask that you take all of their concerns into consideration as examples of how the review could be improved.

·

Basic reporting

I commend the authors for giving an overview over the literature on peripheral-physiological and neural correlates of flow during video gaming. I believe that psychophysiological methods represent a promising approach to both research on flow experiences in particular and on gaming experiences in general. Given the in part contradictory findings and different research methods and designs employed so far, I think that this literature review will greatly benefit the field by summarizing our current knowledge and state of the art. Considering the various possible applications of flow research in health-related areas and the focus on psychophysiological methodology, I further believe that this article will be of high interest for PeerJ’s readers. In terms of English language and grammar, the manuscript satisfies professional standards required by PeerJ and can be allowed for publication. However, I would like to see a major revision of the manuscript’s structure before publication as I found it at times confusing to read and hard to grasp the main information from the text. Therefore, I will now make detailed recommendation on how to change the reporting:
1. When giving the definition of flow, processing fluency is not mentioned; however, apart from attentional absorption and intrinsic enjoyment, processing fluency should be considered a key dimension of flow (Csikszentmihalyi, 1975), from which many psycho- and neuropsychological hypotheses can be deduced. Therefore, the role of processing fluency during flow needs to be made explicit in lines 38-40, 46, and 120-124.
2. The flow factors given in lines 45-50 should be ordered into antecedents and components: The main antecedent for flow is the optimal balance of skills and challenges (theory: Csikszentmihalyi, 1975; correlation: e.g., Jackson & Marsh, 1996; experiment: e.g., Keller & Blomann, 2008; meta-analysis: Fong, Zaleski, & Leach, 2015); therefore, I would like to see a small paragraph dedicated to it (maybe directly referencing the flow channel model), to the experimental approach it inspired (lines 82-90), and to personality traits that modulate its causal relationship with flow (lines 90-98). This way, a single paragraph would summarize the theoretical background for the experiments you are about to report. I also think the statement from lines 468-469 should be part of this introductory paragraph, so the reader knows right away about potential methodological limitations.
3. Concerning the flow components, I suggest you group 2), 3), 7), and 8) together as they are all related to absorption. In contrast, 4), 5), and 6) are all related to fluency. At this point, it should be clarified that 4)-6) can both be regarded as activity characteristics and thus antecedents of flow, and as subjective perceptions during flow and thus flow components. Please also refer to Jackson and Marsh (1996), when giving these flow components since the differentiation between 4)-6) stems from their work. Also, include intrinsic enjoyment as another flow component, which is sometimes conceptionalized as an antecedent or a consequence of flow, too. This is important since some flow questionnaires measure intrinsic enjoyment as a flow component (e.g., Jackson’s flow scale).
4. In regard to experimental approaches in flow research, both the dynamic matching of skills and challenges (which you refer to in lines 66-68; e.g., Keller & Blomann, 2008) and pre-testing skills to assign matching challenge levels (e.g., Moller, Csikszentmihalyi, Nakamura, & Deci, 2007) should be mentioned in addition to creating a medium challenge level (lines 85-90). I would also recommend to at this point briefly state the possibility of having more than three experimental conditions as some of the experiments you are about to report feature another condition (immersion; autonomy), which might confuse the reader later on. Moreover, I know the terminology is not used consistently in flow research, but I think the reader would greatly benefit from you establishing standards for your paper, i.e. stating something along the lines of “When the game’s difficulty is set to a high level, the gamer is assumed to experience stress, frustration, or anxiety rather than flow. In the following, we will refer to the corresponding experimental condition as the overload condition when describing the results of a flow experiment.” Please also make sure that you then use consistent terms throughout lines 120-527.
5. I would also like to see a paragraph on flow measurement to early on give the reader an idea of how flow was assessed in the studies that you are about to report. In this paragraph, you should mention the ESM and flow questionnaires as direct, but retrospective flow measures as well as the dual task approach and physiological indicators as indirect, but non-disruptive flow measures. At this point, the limitations of assessing a state with retrospective self-reports should also be clearly stated, making the case for neural or peripheral-psychological indicators. Thus, please form a new paragraph on flow measurement from lines 69-73, 103-108, and 469-479. I would also prefer more references for flow questionnaires (line 73), ideally by adding a corresponding column to Tables 1-4.
6. Lines 51-64 need to be marked more clearly as speculation or more evidence on the 'beneficial nature of flow on psychiatric symptoms' needs to be given. Also, a clearer explanation of the reasoning behind flow being beneficial in a clinical context should be provided. Otherwise, I would just leave it out altogether.
7. In line 110, you refer to this manuscript as the first review of this kind. However, I am aware of at least one comparable review (Knierim, Rissler, Dorner, Maedche, & Weinhardt, 2018). Whereas I think, your work still adds valuable information and provides an important update, the other review (maybe there is even more?) should be referenced properly and the added value of your literature review should be explained.

Experimental design

I am confident that the manuscript cites relevant sources correctly and fulfills the methodological and ethical standards required by PeerJ. However, I believe improving the organization of subsections and of the overall structure of the manuscript is necessary to give the reader a clearer overview of the topic. Moreover, to ensure comprehensive, unbiased coverage, I strongly suggest to revise the methodological approach. Please consider my recommendations to improve the results section:
1. In regard to the method used to collect papers on peripheral-physiological and neural correlates of flow during video gaming, I strongly suggest to follow PRISMA guidelines (http://prisma-statement.org, see checklist and flowchart). This would also help the reader to evaluate the study-selection strategy. Specifically, the combinations of keywords used should be made explicit (e.g., ‘neural OR correlates OR brain activity AND flow’) as well as all eligibility criteria. I am also not convinced that Google Scholar can be used to obtain a comprehensive overview and therefore strongly encourage to further use PubMed and Web of Science. Maybe more specific keywords pertaining to certain methods/indices and flow would also help to make sure all relevant papers are found. Just to give you an idea, from my own literature research I can name papers on flow and psychophysiology that you have not included in your literature review: Kivikangas (2006), Drachen et al. (2010), Gaggioli et al. (2013), Bastarache-Roberge et al. (2015), Colzato et al. (2018), Plotnikov et al. (2012), Peifer et al. (2014b), and Tozman et al. (2017). I believe it is crucial to revise your literature review to ensure that no invalid conclusions are drawn due to the lack of certain relevant papers.
2. Whereas I find structuring your report into a subsection for peripheral-physiological and neural correlates of flow very helpful, I think the subsections themselves need to be structured more clearly. I suggest to start with a paragraph summarizing theoretical assumptions about flow and the corresponding measures like you did in line 266-280 for the neural subsection. For the neural subsection, you mention the hypofrontality hypothesis (lines 273-274, 358-359) as well as synchronization theory (lines 275-280); however, I would in the same paragraph also mention the notion of effortless attention (lines 568-571), peak performance (lines 130-131, 378-380), and the flow components loss of self-awareness, attentional focus, and intrinsic enjoyment (please make sure to use a consistent terminology here as you also call the latter concentration or intrinsic reward at times), since they, too, inform about potential neuronal correlates of flow. That would help the reader understand which neural correlates are to be expected before learning the actual results. Please note that the link between flow and peak performance should be brought up and explained (!) already in the introduction, preferably when you write about processing fluency and optimal challenge. As of now, it comes a bit out of nowhere when it is used to explain certain results, which is a good example of what I mean by 'confusing to read'.
3. Similarly, I would recommend expanding on the theoretical background of peripheral-physiological flow correlates in a separate paragraph before presenting results. Please include the inverted U-shaped hypothesis by Peifer et al. (lines 181-184, 539-544), De Manzano et al.’s conceptualization of flow as a state of positive affect and high arousal (lines 160-162; you call it effortless attention but as far as I know De Manzano speaks of arousal and Bruya (2010) speaks of effortless attention, so that should not be confused), and the notion of co-activation of the parasympathetic and sympathetic system during flow (Ullén, De Manzano, Theorell, & Harmat, 2010) in this paragraph, as well as briefly effortless attention and peak performance as these two will reappear in the neural subsection. At this point also comment on assumptions about flow and high arousal/stress (lines 124-127), but please clarify right in the beginning that this is presumably highly dependent on the activity context. You bring up task-dependency only in the neural subsection in line 324, but it applies to all psychophysiological results and should be emphasized right in the beginning!
4. In addition to presenting the theory in a clearer manner, I would also appreciate a clearer presentation of the indicators themselves. Even though PeerJ’s readers will most likely be familiar with psychophysiological indicators, you present a broad variety of methods from eye-tracking to tDCS in this literature review, so readers may not know about all of them. As of now, you explain some indicators when you present corresponding results (e.g. LF-HRV, lines 146-149), but I would like to see this done consistently for all measures employed (peripheral-physiological subsection: HR and HRV, EMG, cortisol, respiration, EDA, and eye-tracking; neuronal subsection: fMRI, perfusion MRI, fNIRS, tDCS, and EEG). Please make sure to refer back to the theory paragraph when you introduce the measures (e.g., cortisol, HPA-axis, stress, Peifer’s hypothesis), so that the reader knows right away how each measure can be related to flow experiences. I would suggest writing a separate paragraph on all indicators and placing it after the theory paragraph, before all results are reported, or writing a short disclaimer for each indicator before presenting the results for this indicator.
5. Please also reconsider the organization of your report of results: You seem to organize the studies according to when they were conducted; however, given the many different methodologies employed, I would recommend to organize them according to a) the indicators assessed or b) the theoretical background. Depending on which one you chose, the other one should be used to form subsections within the paragraph (e.g., if you choose to go by indictors, first present all evidence this indicator provided on for instance the effortless attention hypothesis, then on the arousal hypothesis, and so on… or the other way around. But keep this structure throughout the report. This is crucial as for instance in line 179-180 you draw a conclusion from the results you just reported and later on state that other results contradict this conclusion; however, it should be made clear within the same sentence or at least paragraph whether a conclusion can be considered valid or not as readers who only scan the text could otherwise pick up wrong information.) This also means that some studies might be brought up several times as they include different indicators or hypotheses – in that case the method should be explained in detail the first time and only briefly referred to from then on. For the sake of clarity, though, it would be very convenient for the reader to get all the information we have on a certain indicator or hypothesis in one single paragraph, instead of having to switch back and forth. Similarly, when you report EEG results, I would like to have different paragraphs/subsections dedicated to results on each frequency band.
6. Please carefully check lines 120-527 in regard to how you report the studies’ methodology. This should always include the name of the game stimulus, the experimental design/conditions or a statement that the study was correlational, and ideally the name or information on the flow measure (or its proxy). In doing so, you can also refer to your tables. In case no proper flow condition or flow measure was used, this should always be pointed out and briefly discussed to inform the reader about potential limitations.
7. I found that for some studies you do a very good job in relating results to hypotheses (e.g., lines 312-314, 430-436), but please really spell out the theoretical background (e.g., name the flow component that can be used to explain the result). Moreover, make sure that you do this for each of the reported studies in a similar way. I also really liked the format in which you presented the link between a theoretical construct and a physiological structure in lines 521-524 (putting the latter in parenthesis behind the former), so maybe use that more thoroughly.
8. I believe that certain results need clearer explanations on your part: the contradicting results on flow and arousal levels (which you also sometimes call tension or stress – again please use a more consistent terminology; lines 192-199); the meaning of LF-HRV (lines 146-147, 563-564; you do not bring up the substantial evidence that this may not be a direct indicator of sympathovagal balance after all except for one sentence in the discussion!; e.g., Reyes del Paso, Langewitz, Mulder, Van Roon, & Duschek, 2013)); flow condition/measure proxies and their limitations, see above (please be very careful with the wording here as for instance line 386 could be misunderstood as you confirming that high performance works as a proxy for measuring flow with a questionnaire).
9. The concept of immersion is brought up in the report of certain studies (lines 150, 400-402), but is never clearly defined (with references no less) and contrasted to flow. I believe this concept is still way more prominent than flow in virtual reality and gaming research, so I would like to actually see a small paragraph in the introduction dedicated to it and to its relationship to flow. This will help gaming researchers to connect with your paper and readers in general to understand studies that distinguish a flow from an immersion condition.

Validity of the findings

I believe that the conclusion drawn in this literature review are well developed and supported. I only have minor recommendations to clarify parts of the discussion:
1. I would suggest that you rename and restructure the subsection of your discussion according to the main hypothesis that will now be outlined in the theory paragraphs beforehand: arousal, effortless attention, hypofrontality, intrinsic enjoyment, and loss of self-awareness. This might require some changes to the structure as for instance effortless attention refers to your discussion of both findings on mental effort (lines 536 onwards) and attentional control (lines 595 onwards). I am certain that this will help the reader evaluate the in part contradictory findings.
2. I would like to see additional brief subsections on methodological conclusions: In lines 551-559, 679-681, 639-656 and 668-691 you nicely discuss the limitations and applications of certain methods in this research area. I believe that these are very important take-home massages for future studies, so these considerations should be expanded on and stressed by dedicating an entire paragraph of the discussion to them. Also, please make recommendations for future research more explicit. Since task-dependency seems to play an important role for this line of research, it should be discussed in some detail.
3. Similarly, one of the main conclusions I would draw from the research so far is what you state in lines 580-583 in connection to task-dependency. I think that this deserves a separate paragraph since it affects what can be considered a tangible research goal for future studies. I would also like to see it discussed more: What does that mean for psychophysiological flow and gaming research? If there probably is no such thing as a psychophysiological standard pattern for flow, why still study flow with psychophysiological methods? You may also want to consider Knierim et al.’s conclusion that only a co-activation of the parasympathetic and sympathetic system might serve as an objective indicator for flow experiences before completely debunking the possibility to find standard flow patterns, or come up with similar conclusions of your own.
4. I have gotten the impression that at times flow and absorption need to be differentiated more clearly. In lines 590-594 you discuss this in relation to automaticity and in lines 664-670 indirectly in relation to attention. However, I think that it needs to be clarified to the reader that high automaticity (which I for the sake of consistency would call effortless attention) can be achieved also under less than optimally challenging condition in overlearned tasks. Similarly, attentional focus can be high also with high challenges if a person is motivated and purposefully concentrates on the task at hand. This should be taken into consideration when discussing corresponding results. In how far does this for instance represent a potential limitation for dual-task designs? I would also like to see it discussed in general, maybe also in relation to immersion: Can we differentiate flow from absorption or immersion in terms of underlying physiological patterns?
5. The usefulness of performance as an indirect measure for flow, which is discussed in lines 670-679, should be marked more clearly as speculation. Ideally, I would also appreciate more references on the relationship between performance and flow, specifically in the field of computer gaming. Please keep in mind that the most straightforward conclusion is still that performance will be better for easier tasks, whereas flow requires a challenging task. The seemingly paradoxical thought that flow is related to peak performance needs to be explained properly (also already in the introduction, see above).
6. I think the conclusion presented in lines 582-583 needs clarification. I find the wording “to some extent” very vague without further explanations.
7. A minor observation, but please do not refer to flow as a feeling (line 664), but consistently as an experience or state.

Additional comments

1. Please note that in line 582 it says Peifer (2012); I believe it should be Peifer (2014).
2. In regard to your tables, I was wondering why you sometimes use parenthesis when giving the m/f behind the sample size to indicate the sample’s gender distribution and sometimes you do not. If there is no reason to do this, I would prefer a more consistent style.
3. Similarly, in your tables you sometimes indicate missing information with a blank space or with a dash (two dashes actually – maybe make it one longer one instead). Please stick to a consistent style.
4. When you summarize conditions under design, you sometimes use capital letters in the beginning and sometimes not. Please also again consider using a consistent terminology whenever possible (e.g., ‘Overload’ instead of ‘Hard’, ‘Boredom’ instead of ‘Slow’, etc.). Sometimes the ‘/’ inbetween conditions got shifted to the beginning of the next line, which should be avoided.
5. You use both ‘and’ and ‘&’ in your tables, but sometimes the ‘&’ is placed in the beginning of a new line. Either spell out ‘and’ consistently or if you use ‘&’ make sure that it does not shift into the next line as this should be avoided in written English.
6. When giving the time frame under design, you sometimes write out ‘x minutes gameplay’ and sometimes just ‘x minutes’; please use a consistent style.
7. At times, there seem to be free lines or broader spacing in your tables; please stick to one line spacing throughout all tables.
8. Beware that I am not a native speaker myself, but I am fairly certain that when giving time frames hyphens and ‘long’ are not used (e.g., ‘2 min’ or ‘2 minutes’ instead of ‘2-min-long’). Please revise your tables accordingly. Also, make sure you stick with either ‘2 minutes’ or ‘2 minute’.
9. When listing more than two measures, a comma should be used before the ‘and’. Again, revise your tables accordingly, please.
10. Table 1 seems to be missing the study of Mauri et al. (2011), which is cited in the text. Please make sure that each study is both described in text and appears in the corresponding table!
11. In the table note for Table 1, I would like you to add explanations for the abbreviations fMRI, fNIRS, and NIRS since they appear in this table. I would also like to encourage a more consistent terminology even though I am aware that in the papers themselves different terms are used; you might consider using one term in regard to cortisol/cortisol level and EDR/GRS, EDA/GSC (and then of course also use it consistently in case it appears in another table). One minor formal observation: there is a blank in the wrong space between the GSR and BP explanations.
12. In the table note for Table 2, you do not give explanations for the abbreviations fMRI, fNIRS, ECG, EDA, NIRS, PET, and tDCS. I believe each table should be understandable in its own right, so that a full explanation of all abbreviations is required for each table. Please also note that you write both ‘FNIRS’ and ‘fNIRS’, but it would be preferable to stick to one style of abbreviation.
13. In the table note for Table 3, an explanation for the abbreviations EEG is missing. Please also again consider using consistent terms for electrodermal indices.
14. In Table 3, there is a spelling mistake for ‘seconds’ when Wolf et al.’s study is presented.
15. In Table 4, you start using a ‘+’ to add the dual task; I think it is a bit unfortunate to use a mathematical sign in a non-mathematical way, so maybe just stick to either using ‘and’ or ‘&’ (but consistently please, also in regard to the other tables).
16. The table note for Table 4 is missing; however, again all abbreviations used in the table should be explained.
17. One more minor observation: the font changes in Table 4 when Huskey et al.’s game is described.

Reviewer 2 ·

Basic reporting

I have appreciated the effort of the authors to write a comprehensive review about the physiological and neural correlates of flow experience (while playing videogames). Authors stated that this is the first review include the studies about physiological and also neural correlates of the flow experience. This is maybe true but I have several questions and problem about about the concept and the purpose of this review. In my opinion neither the studies of the physiological nor the neural correlates of the the flow experience have not discussed in deep way. In my general opinion, this review is still not ready to publication because of several weaknesses and I can not see what this review add to the field. I suggest major revision or re-submission to the editors for this work.

Experimental design

The selection method for the review is poorly reported. Authors should provide detailed description about the combination of the keywords, how many article were find, the inclusion and exclusion criterias to final selection of the articles have not reported in the review. I suggest the author to report their review selection method in details.

Validity of the findings

Validity of the results
The authors reported several studies on the field without applying their deep critical opinions about the results and suggestions for future research. Authors suggested poor conclusions to summarize the studies about the physiological correlates of the flow experiences e.g. ”results are inconsistent…” OK, all the experts on the field can agree with this. What is major problem to find specific patterns for the underlying physiological and neural mechanisms of the flow experience based on the current literature? What are the author’s suggestion for future research to improve the field? I think some book chapters discussed the problem in more detailed and comprehensive way and this knowledge should be integrated into this paper also (Tozman & Peifer et al. 2016; Keller et al. 2016). It was interesting to read the studies about the neural correlates of the flow experience. I have no knowledge any other review artcile on this field. On other hand, the results was not discussed in good details and I would like read more about the different approaches and studies on this field. In addition, I don’t understand the concept and purpose of the review. If the authors aimed to review the physiological and neural correlates of the flow experience in one article I would appreciate some discussions about the connection between two fields. What we can learn to understand more about flow if we scope together the psysiological and neural correlates of the flow experience? What are the future directions in flow research to understand the connections between the neural correlates and the specific peripheral-physiological patterns of the flow experience??

Additional comments

I suggest to re-think the concept of the review. Maybe it would be enough if the authors just focus on the neural correlates of the flow experience with some critical review on the existing studies and add some suggestions how the reserach should be developed. This would really increase the validity of this paper and may fill some gaps on our field. If still the authors want to keep the peripheral-physiological part of the paper I suggest that make deep discussion about the question why the existing results are inconsistens.
Some minor comments
However most of the flow studies was made on computer gaming there are several studies was made on other tasks e.g. De Manzano 2010, Peifer 2014. Based on the title of the paper the authors focused on studies on computer gaming and flow. This problems relates to confusion about the review selection method and also the concept of the article.
Authors only introduce an eight dimension model of flow. I suggest to introduce the nine dimension model by Csikszentmihalyi 1990 that was also used in FFS-2 scale (Jackson & Eklund, 2002) and add autotelic experience (enjoyment) as a dimension. Athuors metioned that in several context in the text ejoyment is an important factor for computer gaimg in connection with flow.
References:
Tozman & Peifer (2016) Experimental Paradigms to Investigate Flow-Experience and Its Psychophysiology: Inspired from Stress Theory and Researc In: Harmat, L., Orsted, F., Ullén, F., Wright, J., Sadlo, G., (eds): Flow Experience: Empirical Research and Applications. Springer, 2016.
Keller, The Flow Experience Revisited: The Influence of Skills-Demands-Compatibility on Experiential and Physiological Indicators. In: Harmat, L., Orsted, F., Ullén, F., Wright, J., Sadlo, G., (eds): Flow Experience: Empirical Research and Applications. Springer, 2016.

---

## Round 0.2 · Minor Revisions

Many thanks for your extensive revisions addressing the reviewer's concerns and suggestions. Overall the manuscript is in quite good shape. I am primarily sending it back to you so that you have an opportunity to correct a few small issues before acceptance (particularly as PeerJ does not perform additional copyediting).

As you can see below, one of the reviewers has taken the time to provide you with extensive additional advice for edits. However, I do not expect you to individually address or incorporate all of these comments and requests in your resubmission: I have included them for your information, so that you may identify and correct factual and grammatical issues where they exist, and make stylistic changes where you think it will increase readability.

I do ask that you _definitely_ address the following issues in your revision:

1) Address reviewer’s concern about processing fluency.

2) ln. 129-130: While I understand the desire to increase consistency across studies, doesn't the naming of conditions as “boredom” and “anxiety” pre-suppose the mental states induced by these conditions? I might suggest that you consider using more neutral names that reflect task difficulty (or difficulty-skill balance) rather than the presumed mental/physiological state.

3) ln. 488: the amygala is not part of the typical definition of the DMN, as far as I am aware.

4) ln. 823: reduced activity in AMY _likely_ reflected decreased emotional arousal

5) give the manuscript another pass for grammatical and typographical errors. I have included my own highlights on the manuscript for spots where I identified issues. Some of these may overlap with those identified by the reviewer.

I will not send your revision out for re-review; your revision will be reviewed only by me.

·

Basic reporting

First, I would like to thanks the authors for extensively revising their manuscript according to both reviewers’ recommendations. I believe that the manuscript greatly improved as it now shows a clearer structure, a more in-depth discussion of both theoretical assumptions and observed findings, and a more comprehensive review of the literature. Therefore, I would only like to make some minor final remarks and recommendations before the manuscript can be published:
1. I noticed some grammatical errors and spelling mistakes throughout the manuscript, which is why I would recommend to have a native speaker proof-read its final version before publication. Examples: leave out ‘either’ (line 100); repeat ‘on’ after ‘as well as’ (line 162); use ‘for the creation of a flow state’ or ‘for creating a flow state’ (line 163); spell U-shaped either consistently with an uppercase or lowercase U (e.g., lines 207 & 211); ‘A significant effect was neither found for…, nor for…’ (lines 224-226); ‘noted’ instead of ‘considered’ (line 256); ‘associated with’ (line 321); ‘capacity’ (line 340) does not seem to be the right word (do you mean the skill to correctly judge the appropriate difficulty level for oneself or the opportunity to pick a difficulty level oneself?); plural ‘flow scores… were’ (lines 413-414); ‘flow as measured by…’ (line 430); ‘a mental arithmetic task’ since the task has not been described beforehand (line 464); ‘higher left temporal cortex alpha activity, which reflects decreased cortical activity in this region, is associated with improved performance, as it represents reduced internal verbalization and increased visual-spatial processing in the right hemisphere’ (lines 531-534); ‘a 7-second video clip’ ‘table tennis’ (line 549); ‘was to be expected in the more difficult condition’ (lines 574-575); ‘which increase with increasing difficulty’ (line 576); ‘the flow and anxiety condition’ (line 579); ‘as compared to’ rather than ‘than’ (line 580); leave out ‘alone’ or replace with ‘on its own accord’ (line 620); ‘playing in 2D’ (line 666); ‘The observed co-activation… aligns’ (line 756-757); ‘flow was found to be associated with’ (line 759); ‘a purely effortful mental experience’ (line 766); ‘practice’ instead of ‘exercise’ (line 789); ‘mind-wandering’ (line 801); ‘body- and self-awareness (line 807); ‘role’ (line 163 and 819); ‘some studies…, other studies…’ (lines 832-833); ‘in which’ instead of ‘where’ (line 833); leave out ‘of’ (line 840); ‘neural activity’ (line 849); ‘…, given the different physiological and cognitive demands of the flow-inducing activity, …’ (lines 850- 851). Moreover, ‘e.g.’ is normally followed by a comma and the gerund is preceded by one (e.g., lines 885-886 ‘…, reporting medium levels of performance’), and an enumeration of three or more things requires a comma before the ‘and’ (e.g., line 122); please make sure to put these commas throughout the manuscript. Also, check the blanks between words as you tend to sometimes either put too many (e.g., lines 236, 267, 486, 498, 513, 649, 660, 742, 841, 886, 892) or none (e.g., lines 44, 469, 470, 486). There are also some stylistic mistakes: In line 355, there is a semicolon that does not seem to make sense; FKS in line 91 is underlined for some reason; the dash once comes with blanks and sometimes without. I do not claim to have found everything, so I strongly recommend the authors to also proof-read their manuscript in terms of formal and stylistic requirements once more.
2. I have some remaining issues with the introduction: ‘Why do we seek activities that make us happy?’ does not inspire interest in the reader as the answer is in the question; I would suggest to just leave it out (line 37-38).
Further, I believe there is still a misconception about processing fluency as well as about antecedents, components, and consequences of flow reflected in the text. Processing fluency is a dimension of flow, just like absorption and enjoyment (i.e., broader than the nine components), and is directly reflected in some of the flow components such as sense of clear goals and immediate feedback. Thus, there is no ‘state termed processing fluency’ (line 40-41), but processing fluency is part of the definition of the flow state and should appear together with enjoyment and absorption in lines 38-39. ‘Stemming from intrinsic motivation’, however, in my opinion does not necessarily belong in the definition itself, as intrinsic motivation is an antecedent (and maybe also a consequence) of flow – therefore it should be mentioned later on together with the first three key components (not dimensions!) as potential antecedents in lines 54-55. Importantly, each antecedent of flow (e.g., clear goals when starting the flow activity) at the same time serves as an experiential component (e.g., the perception of clear goals during flow in this activity). To make this clearer, I would reword lines 54-56 as ‘Whereas all nine key components reflect the subjective experience during flow, the first three (balance between skills and challenges, clear goals, and immediate feedback) can in a more objective sense also be understood as activity-related preconditions for flow. Moreover, in relation to the person, intrinsic motivation for the activity can facilitate a flow experience (Csikszentmihalyi, 1975). Optimal skill-challenge balance, though, is considered to be the main antecedent for entering a flow state…’.
3. I would also suggest to re-structure the introduction to make it easier to follow for the reader: After you introduce the flow channel model and potential modulators of the relationship it predicts between flow and skill-challenge balance (lines 59-75), explain how this theoretical framework can be relatively easily implemented in the context of computer gaming (‘…could facilitate the propensity to experience flow. – new paragraph – Despite such potential modulators, the relationship between skill-challenge balance and flow postulated in the flow channel model affords an approach for experimentally manipulating flow by presenting different challenge levels. Even though inducing flow under controlled laboratory conditions has been considered difficult (reference please!), game-based paradigms are especially promising as a) the difficulty level of a game can relatively easily be manipulated to achieve skill-challenge balance, and b) gaming can be considered a typical flow-inducing activity’; then go on with the reasons why video gaming is a great flow activity from lines 104-112 and 132-133, and follow up with the manipulation approaches from lines 113-118). Please make sure to include all three approaches, namely also the medium challenge level-approach (Rheinberg and Vollmeyer, 2003), in lines 113-117 – and then move lines 124-126 up as an example (‘For instance, Rheinberg and Vollmeyer…’). In the next paragraph, you can then give three interesting areas of application for experimental flow research with computer games: a) understanding peak performances, b) potential use of flow in therapeutic settings, and c) maximization of game enjoyment. Thus, put lines 75-84, lines 131-148 and lines 149-153 next, respectively (with appropriate connecting passages of course). Then, you can come to the methodological challenge this research faces and its potential solution in the form of psychophysiological or neural indicators for flow, pointing out why the reader should be interested in what is about to follow in your literature review (e.g., ‘In order to successfully study research questions related to flow and performance, therapeutic use, or gamer experience, the experimental paradigm must be accompanied by a valid measurement of the flow state.’; move lines 85-103 here – how flow used to be measured with self-reports, how this is problematic and warrants additional psychophysiological and neural measures; now also briefly mention the dual-task approach from line 156-158 and in doing so, please spell out ‘dual-task approach’). And then you can end with lines 158-171.
4. There are some minor unclarities left in the introduction: ‘Autotelic experience’ (line 52) – this term needs to either be explained (Greek: ‘auto’=self, ‘telos’=goal) or reworded as ‘intrinsic enjoyment, which is more self-explanatory. In lines 67-68, it is ‘activities that were subjectively evaluated as important to succeed in’ ¬– this way, the obtained result (low demands/flow) becomes more understandable. The FSS (lines 90-91) was developed in the context of sport, but can be and is often applied to other flow activities, just like the FKS. The FKS is called Flow Sort Scale in English and should therefore also be abbreviated to FSS and not FKS. The FQ (lines 668-669) is missing in this section and should be introduced with the ESM as it was originally developed for ESM studies (lines 85-88). Effectance motivation might as well be explained using the term ‘feeling of control’ instead of ‘empowerment’, pointing out the link to this flow component. The TOJ (lines 143-148) needs a bit more explanation, since the loss of a sense of time during flow would suggest that a flow experience actually diminishes the performance in a temporal judgement test. Similarly, the reasoning why autonomy is suggested as an important determinant of flow needs to be briefly given (lines 119-120, lines 340-341). And I am painfully aware of the difficulties to distinguish flow and immersion in video gaming, but since the term immersion is later brought up, there needs to be one or two comprehensive sentences about this (‘Some studies (ref.) implemented an additional immersion condition, which was aimed at transporting the player’s mind into the virtual world by manipulating factors like graphics, sound, and gameplay (ref.). Immersion is an important concept in VR and computer gaming research and seems closely connected to flow in these activity contexts, as both pertain to a pleasant, absorbed mental state common in gamers. However, flow and immersion can be distinguished due to subtle structural differences (for further information on this, please see Michailidis, Balaguer-Ballester, & He, 2018).’.

Experimental design

I am very happy to see that the authors put in the additional effort to extend their literature research to other databases and to revise the presentation of their inclusion criteria. The methodology is now much clearer and more convincing. I also find the new structure of the presentation of the findings much easier to follow for the reader. Again, I would like to commend the authors for the successful revision, and only have minor remarks to add:
1. I believe some section need a bit more clarification: In line 182, the term ‘exergames’ may need a short explanation and the confounding factors could be accompanied by examples (e.g., movement effects, processing of social interaction, …), in order to explain to the reader why the 35 papers were excluded, to be as transparent and detailed as possible with the exclusion criteria; in lines 204-205, I would write ‘that the experience of challenge during flow induces a certain amount of stress’ (lines 204-205) since ‘stress’ and ‘challenge’ cannot be used interchangeably in some theories; in line 206 and 207, I find ‘fast/slow response system’ to be better terms, respectively; in line 228 and 556, please leave out ‘emotional’ as flow is not an emotional state; in line 236, you state that flow was assessed in the study and later go on to say that flow was actually not assessed with a questionnaire (line 239); you may want to say that there was a flow, immersion, and boredom condition, but no measurement of self-reported flow?; in line 445, please write ‘a first person shooter (FPS) game’; lines 447-449 could use an example to clarify what is meant by ‘game events that contribute to factors described by Csikszentmihalyi’; in line 433 the name of the FPS game, in line 419 the name of the videogame, and in line 656 the name of the flow scale is missing.
2. I find it a bit adverse to first give interpretation of the HRV indices and later on discuss those interpretations. I would be upfront with the controversial aspects here: ‘To investigate the relationship between flow and ANS activation, cardiovascular measures are oftentimes assessed, given that heart rate variability, which pertains to fluctuations in the time intervals of adjacent heartbeats, is traditionally seen as an indicator to dismantle PSNS and SNS activation. Thus, high-frequency HRV can be considered a direct measure of parasympathetic activation, whereas low frequency HRV has long been interpreted as reflecting sympathetic influences, rendering the HF/LF ratio an indicator for autonomic balance between SNS and PSNS. However, it is important to note that this interpretation of LF-HRV and LF/HF ratio is controversial…’ and follow it up with the information from lines 293-301. After this clarification, you can go on to report De Manzano et al. (2010) and later Chanel et al. (2011).
3. Sometimes it is unclear how certain constructs related to flow were measured, given that they could be measured both with self-reports and physiological indicators. For instance, in line 277 do you mean self-reported positive affect and effortless attention? Same for line 321: How was the decrease in mental effort assessed? In line 416 you mean the last 30 seconds of the flow condition, and in line 478, you mean the flow score? Similarly, in line 500, do you mean in the flow condition, and in line 501, correlations between the flow score and DMN activity? What is the attention score referred to in line 595? In line 868, I assume you mean ‘…does not guarantee that a person will enter a flow state in the flow condition.’. In line 489, you mean a flow score – or is the flow index something else? And in line 615, what do you mean by ‘moments of flow’ since as of now flow cannot be captured with high temporal accuracy?
4. I would re-structure and re-word certain lines to clarify their meaning: In lines 200-201, I would suggest. ‘In contrast to the notion that the attentional focus during flow is essentially effortless, Keller and colleagues (2011) argued that flow involves considerable mental effort due to the high degree of involvement along with the challenging nature of the task.’; in line 234, ‘EDA measures increased with the game’s difficulty level, indicating heightening arousal’; in lines 323-329, ‘The different findings show that during… participants may not perceive as much stress…. It should also be noted, that Keller’s study involved a relatively small number of participants and operationalized flow through the flow condition, without actually measuring self-reported flow experiences, which might affect the validity of their findings.’ (On a side-note, if this operationalization is used, the analyses is most likely not correlational.); in line 401, ‘the activity of the DMN is thought to decrease’ since you are still stating assumptions and not findings at this point; in line 459-461, ‘It should be noted, though, that in the second study, the subjective flow experience was assumed to be elicited by the flow condition, but this assumption was not checked using a self-report flow scale.’; in line 544, either say ‘once again’ or refer to the other studies; in line 491, ‘, indicating less self-related processing,’; in lines 556-560, ‘Chanel et al. (2011) tried to classify their three different challenge-skill conditions in a game of Tetris according to EEG signals. Even though their results indicated distinct theta power between conditions in some electrodes, no precise link to states of boredom, flow, and anxiety could be obtained.’; In line 843, I guess you mean the ‘flow channel model’; in line 856, leave out ‘would’ and go on to say ‘FPSs are three-dimensional games, in which spatial navigation is crucial, since at any given time, ….’ and delete the sentence (line 860) which merely repeats the part about spatial navigation; in line 868-869, since you give a recommendation, say ‘The subjective experience of flow in the flow condition should be assessed using self-report flow scales in future studies, to determine whether the participants were able to enter a full flow state or not.’.
5. I noticed that there is some repetition in the text (for instance, it is stated on multiple occasions that a flow condition does not guarantee a flow experience). Moreover, when describing a study, there is often a more general sentence first (e.g., line 412, ‘playing a video game’), followed by a more specific sentence (e.g., line 413 ‘while playing Tetris); this information could be presented in a more concise form in just one sentence. Finally, brackets are often used to insert full sentences (e.g., line 431-432); however, for fluent reading it is advisable to use brackets only to give short key words and to use a subordinate clause when more text is needed. Please revise the text once more according to these opportunities for improvement.

Validity of the findings

I am confident that the revised, more in-depth discussion of the findings together with explicit recommendations for future study designs makes this literature review an interesting read for PeerJ’s readership and clearly shows its added value to the field. In addition to some minor language- and grammar-related issues I have already brought up in the other sections of the review, my only remark here is, whether the ‘joyous state of focused attention’ should not be re-named as a ‘subjectively effortless state of focused attention’ in order to set it apart from a ‘state of positive valence and heightened arousal’. Other than that, I find the discussion section very well thought-out!

Additional comments

At this point, I have to stress the importance of proof-reading a manuscript multiple times to ensure a consistent and correct style; I myself have experienced that a manuscript can get rejected due to formal mistakes. I have put a lot of effort in trying to find these mistakes for you, but I do not claim to have found all of them, so please check again. A good example for persistent inconsistencies and formal mistakes are the tables:
1. For ‘Experimental type’, you write either ‘FPS Game/game:’ or ‘Game’ with no colon. Similarly, for ‘Design’, you write either ’40 minutes gameplay’, ’10 minutes’, or ‘3 minutes blocks of gameplay’. Stick to one way of reporting. I would suggest ‘(FPS) Game:’ and ‘Boredom/Flow/Anxiety, 10 minutes gameplay’.
2. For ‘Measures’, it happens quite often that the ‘&’ is in the beginning of a line or a comma is missing; to avoid these mistakes, I would suggest simply enumerating all indicators with commas (e.g., ‘EDA, HRV, fEMG’ instead of ‘EDA, HRV, & fEMG’). Same for multiple games under ‘Experimental Design’ – here, only for the dual tasks the ‘&’ should be used.
3. Still, some commas are missing when three or more items are enumerated throughout the tables. In the last column, I would also suggest to only use the ‘&’ when indicators are enumerated, but not for results – in this case, use ‘;’ (e.g., Table 1 ‘Inverted U-shaped relationship between LF-HRV & cortisol level with self-reported flow; positive linear relationship between parasympathetic activation & self-reported flow).
4. Some formal mistakes remain: in Table 1, for Chanel et al. the s in participants is shifted to the next line, there are superfluous blanks in the lines for Leger et al. and for Harmat et al. under ‘Design’, a bracket is missing in ‘77(40f)’, and the enumeration in the last column of Leger et al. should be ‘A, B, & C’. In Table 2, there is another superfluous blank in ‘Moreno et al. (2020)’, ‘autonomy condition’ in the last column of De Sampaio et al. should be lowercase as is flow condition in the other lines, and the enumeration in the last column of Klasen et al. should be ‘A, B, & C’. In Table 3, under ‘Design’ and DeKock et al. the font size changes. In Table 4, the ‘&’ is in the start of the line under ‘Experiment type’ and Castellar et al..
5. Why not put ‘fEMG’ instead of ‘Facial EMG’ (of course explaining the abbreviation below)? Same for ‘RR’ and ‘Respiratory Rate’.
6. It is important to clarify in the last column how the findings were related to flow. Sometimes you write ‘for flow’ and ‘during flow’, sometimes ‘for flow condition’, ‘during flow condition’, and ‘at flow condition’ or ‘in the flow state’, ‘with the flow experience’, and ‘during self-reported experience of flow’. This is unnecessarily confusing. I would suggest you stick to ‘in the flow condition’ whenever the analysis was based on group differences, and to ‘during self-reported flow’ whenever it was based on correlations with a flow score.
7. ‘LF-HRV’ and ‘HF-HRV’ should consistently be spelled with the hyphen.
8. In Table 2, please put ‘Increased activity in the left IFG, left putamen, & posterior cortical regions during self-reported flow; decreased activity in MPFC, PCC, & AMY during self-reported flow’.

---

## Round 0.3 · Minor Revisions

The manuscript is in great shape. I have only a single request before it is ready for acceptance and production.

Regarding the use of the term "processing fluency" in the definition of flow: I think the reviewer has a point here that is not adequately addressed. Part of the problem is that processing fluency is an influential idea in several fields that has developed a quite specific meaning, primarily after the work of Norbert Schwarz and his colleagues. Rather than get caught up in this, I suggest the following edits to lines 37-41, which is more in line with the reviewer’s suggestion:

Csikszentmihalyi (1975) introduced the concept of “flow” or “being in the zone” as an optimal state in which complete absorption in an activity is reached and is accompanied by a sense of enjoyment and apparent ease of processing. Flow is a subjective experience in which actions seem to happen effortlessly, fluently, and almost automatically.

If you agree, please make this change and resubmit the manuscript file and I will accept immediately.

---

## Round 0.4 · accepted · Accept

Thank you for all your work to improve the manuscript. I'm sure many researchers will find this review to be of use.